# Country-scale greenhouse gases budgets using shipborne measurements: a case study for the United Kingdom and Ireland.

Carole Helfter[1], Neil Mullinger[1], Massimo Vieno[1], Simon O'Doherty[2], Michel Ramonet[3], Paul I. Palmer[4] and Eiko Nemitz[1].

[1]Centre for Ecology and Hydrology, Penicuik, UK
[2]School of Chemistry, University of Bristol, UK
[3]Laboratoire des Sciences du Climat et de l'Environnement, CEA-CNRS-UVSQ, Gif-sur-Yvette, France
[4]School of GeoSciences, University of Edinburgh, Edinburgh, UK

*Correspondence to*: Carole Helfter (caro2@ceh.ac.uk)

**Abstract.** We present a mass balance approach to estimate the seasonal and annual budgets of carbon dioxide ($CO_2$) and methane ($CH_4$) of the United Kingdom (excluding Scotland) and the Republic of Ireland from concentration measurements taken on a ferry along the east coast of the United Kingdom over a 3-year period (2015-2017). We estimate the annual emissions of $CH_4$ to be 2.55 ± 0.43 Tg, which is consistent with the combined 2.29 Tg reported to the United Nations Framework Convention on Climate Change by the individual countries. The net $CO_2$ budget (i.e. including all anthropogenic and biogenic sources and sinks of $CO_2$) is estimated at 881.0 ± 137.5 Tg, with a net biogenic contribution of 458.7 Tg (taken as the difference between the estimated net emissions and the inventory value which accounts for anthropogenic emissions only). The largest emissions for both gases were observed in a broad latitudinal band (52.5 °N – 54 °N), which coincides with densely populated areas. The emissions of both gases were seasonal (maxima in winter and minima in summer), strongly correlated to natural gas usage and, to a lesser extent, also anti-correlated to mean air temperature. Methane emissions exhibited a statistically significant anti-correlation with air temperature at the seasonal time scale in the central region spanning 52.8 °N – 54.2 °N, which hosts a relatively high density of waste treatment facilities. Methane emissions from landfills have been shown to sometimes increase with decreasing air temperature due to changes in the $CH_4$-oxidizing potential of the top soil, and we speculate that the waste sector contributes significantly to the $CH_4$ budget of this central region. This study brings independent verification of the emission budgets estimated using alternative products (e.g. mass balance budgets by aircraft measurements, inverse modelling, inventorying) and offers an opportunity to investigate the seasonality of these emissions which is usually not possible.

## 1 Introduction

The United Nations Framework Convention on Climate Change (UNFCCC), kick-started an international political drive to reduce emissions of greenhouse gases (GHG) and stabilise their mid- to long-term impact on the global climate. The focus of the international community over the past 2-3 decades has been on curbing emissions of carbon dioxide ($CO_2$), the most abundant and well-understood GHG, but it is now recognised that emissions of other GHGs such as methane ($CH_4$) and nitrous oxide ($N_2O$) must also be addressed in order to fulfil the goal of limiting irreversible climate change set out under the 21st Convention of Parties (COP21). Reductions in $CH_4$ emissions in particular would be effective in reducing GHGs more quickly, given its shorter lifetime. Annex 1 parties are required to report their GHG inventories annually to the UNFCCC following the guidelines set out by the Intergovernmental Panel on Climate Change (IPCC, 2006; UNFCCC, 2014). Emissions inventories are powerful tools but they intrinsically rely on detailed knowledge of source abundance and strength and they can therefore carry significant uncertainties. For example, uncertainties on the fossil fuel emissions from Europe and North America have been estimated to be of the order of 2% to 5% (Marland, 2012); in contrast, a 1.4 Gt gap in $CO_2$ emissions was reported in China in 2010, which was equivalent to ~ 5% of the global budget (Guan et al., 2012). Because much of its emission is directly

linked to the amount of fossil fuel used, $CO_2$ is the best-understood GHG but, despite this, regional and consequently global emissions budgets are thought to be under-estimated and the uncertainties are increasing due to the growing emissions from emerging economies (Gregg et al., 2008; Gregg, 2008; Peters et al., 2011). By contrast, relatively more of the $CH_4$ emission is mediated by biological processes. After a period of stagnation in the first few years of the 21$^{st}$ century, atmospheric $CH_4$ has

been rising steadily since ca. 2007. This prompted renewed efforts by the international scientific community to identify the drivers of $CH_4$ at local, regional and global scales and reconcile bottom-up and top-down estimates (Kirschke et al., 2013; Saunois et al., 2016). At the global scale, total methane emissions from fossil fuels (from the fossil fuel industry and from geological seepage) have been relatively steady over the past three decades but research indicates that the estimates must be revised upwards by as much as 60%-110% (Schwietzke et al., 2016). Several mechanisms have been proposed to explain the

recent rise in atmospheric methane; these include increases in emissions from microbial sources, which are meteorologically driven and can therefore exhibit substantial inter-annual variability (Dlugokencky et al., 2011; Nisbet, 2016; Schwietzke et al., 2016), a weakening of the hydroxyl (OH) chemical sink strength (Rigby et al., 2017; Turner et al., 2017) and an increase in fossil fuel contributions in the context of a stable OH sink and a downward revision of the biomass burning budget term (Worden et al., 2017). Inventories are thought to over-estimate global emissions and a difference of 130 Tg $CH_4$ y$^{-1}$ was found

between bottom-up and top-down estimates (Kirschke et al., 2013). In this light, it is becoming increasingly clear that there is an urgent need to seek independent validation of the emissions inventories using an integrated range of measurements and modelling activities (Allen, 2016; Nisbet and Weiss, 2010).

The development in recent years of rugged, high-precision spectroscopic instruments (e.g. Peltola et al., 2014) has opened up new opportunities for continuous, in-situ measurements of methane at fine temporal resolution and at relatively large spatial

scales. For example, such sensors have been used in airborne applications to study methane emissions from cities (Cambaliza et al., 2015; Cambaliza et al., 2014; Mays et al., 2009; O'Shea et al., 2014) as well as anthropogenic and biogenic area sources (Hiller et al., 2014; Karion et al., 2015; Karion et al., 2013). Applications of eddy-covariance to measure emission / deposition directly at the field scale are on the rise in a broad variety of environments ranging from agricultural and wetlands (Dengel et al., 2011; Erkkila et al., 2018; Felber et al., 2015; Meijide et al., 2011; Peltola et al., 2015; Podgrajsek et al., 2014; Nicolini et

al., 2013) to urban (Gioli et al., 2012; Helfter et al., 2011; Helfter et al., 2016; Pawlak and Fortuniak, 2016; Pawlak et al., 2016). Finally, networks of tall towers and networks thereof exist around the world to monitor and model methane emissions at spatial scales ranging from country to global (Bakwin et al., 1995; Bohnenstengel et al., 2015; Feng et al., 2009; Feng et al., 2011; Miller et al., 2013; Oney et al., 2015; Rigby et al., 2008; Stanley et al., 2018; Stavert et al., 2018).

The geography of the British Isles at the NE edge of Europe, with dominantly SW winds advecting clean Atlantic air masses,

particularly lends itself to a further approach, namely boundary layer budget measurements of the concentrations in the in- and outflow. This has previously been utilised for airborne boundary layer budget measurements (Fowler et al., 1996; Gallagher et al., 1994; Polson et al., 2011), but these can only provide snapshots of the country emissions for a few individual days.

Extending the concept of these earlier studies, we present three years of continuous observations (2015-2017) of $CO_2$ and $CH_4$

concentrations measured on-board a commercial freight ferry, which connects the ports of Rosyth (Scotland, UK) to Zeebrugge (Belgium) and tracks the East Coast of most of Great Britain. The route of the ferry transects the UK outflow with a time-dependent footprint (Fig. 1), which, combined with typical West-to-East air mass travel time of 11 to 19 hours across the domain (median values for winter and summer, respectively) allows for sub-daily emissions estimations. Furthermore, the three continuous years of measurements also provide an opportunity to study seasonal and inter-annual changes in emissions.

This is to our knowledge the first example of country-scale emission budgets using a mobile platform measuring continuously over several years.

These shipborne measurements formed part of a larger observation and modelling program - the Greenhouse gAs Uk and Global Emissions (GAUGE) project (Palmer et al., 2018) - aimed at determining the magnitude, spatial distribution and uncertainties of the UK's GHG budgets of $CO_2$, $CH_4$ and $N_2O$. In this paper we utilise shipborne observations at the outflow of the United Kingdom coupled with background measurements at the Mace Head site (Ireland) to estimate seasonal and annual budgets of $CO_2$ and $CH_4$ using a mass balance approach.

## 2. Materials and methods

This section describes the measurement systems used at the two experimental sites (Mace Head and ferry) and provides details of the greenhouse gas budget calculations and mass balance approach.

### 2.1 Experimental setup

#### 2.1.1 Shipborne measurements

Measurements of concentrations of carbon dioxide ($CO_2$) and methane ($CH_4$) began in February 2014 on board a commercial freight ferry (operated by DFDS Seaways) which served the route between Rosyth (Scotland, UK; 56° 1' 21.611'' N, 3° 26' 21.558'' W) and Zeebrugge (Belgium; 51° 21' 16.96'' N, 3° 10' 34.645'' E). The route of the ship followed the outline of the English coast on the East side of the UK, which placed it downwind of Atlantic air masses blowing in from the West over Ireland, Wales and England (Fig. 2). The ship completed three return journeys per week and typically operated for 48 weeks per calendar year. The schedule of the weekly cruises allowed for the latitude range to be sampled at different times of day and night as illustrated in Fig.3.

The roll-on/roll-off (Ro-Ro) cargo / container carrier vessel Finnmerchant (former name Longstone; IMO: 9234082; overall length and breadth: 193 m x 26 m) served the Rosyth – Zeebrugge route during the measurement period 25/02/2014 – 15/04/2014. It was replaced by the Ro-Ro cargo ship Finlandia Seaways (IMO: 9198721; overall length and breadth: 163 m x 21 m) on 15/06/2014 and measurement restarted aboard the new vessel.

Air was sampled on the topmost platform of the bow (port side on Finnmerchant and starboard side on Finlandia Seaways) and analysed by a cavity ring-down spectrometer (CRDS; Picarro 1301) housed in an air-conditioned measurement container located on the weather deck. Air was pumped at ca. 50 lpm through 20 m and 50 m of ½'' Synflex tubing at approximate measurement heights 20 m and 30 m a.s.l. (on Finnmerchant and Finlandia Seaways, respectively) and sub-sampled at ca. 10 lpm through 2 m of ¼'' Synflex tube by a secondary pump. The sub-sampling tee-piece was set up as a virtual impactor to prevent moisture and sea salt from entering the CRDS sampling line and the main sampling line was protected from moisture ingress by three water traps arranged in series (at the ambient air inlet point as well as immediately upstream and downstream of the virtual impactor).

The gas measurement system was equipped with a bespoke auto-calibration system controlled by an in-house LabView™ program which also handled the acquisition of data (0.5 Hz) from the Picarro gas analyser, a weather station (WXT520, Vaisala) co-located with the air inlet on the ship's top deck and a GPS (18x series, Garmin) receiver affixed to the roof of the sea container. Calibrations using three gases spanning a realistic range of $CO_2$ and $CH_4$ concentrations ran every 169 hours and lasted 65 minutes in total. The references gases were calibrated by the Swiss Federal Laboratories for Materials Testing and Research (EMPA, Dübendorf, Switzerland) using a Picarro 1301 CRDS. The calibrations scales (NOAA/ ESRL) were WMO-CH4-X2004 for methane and WMO-CO2-X2007 for carbon dioxide. Each gas standard was measured at 1 Hz for 15 minutes and average and standard deviation were derived for the 15-minute period. A 5-minute purge period using the gas standard to be measured was observed before each active averaging period to flush out residual gas and eliminate sample contamination. Each calibration event ended with a 5-minute purge period using ambient air before resuming normal operations. The gas concentration time series were corrected using linear temporal interpolations between calibration events.

Table 1 provides a list of observables; Table 2 summarises the weekly auto-calibration procedure and provides information on the three calibration gases used.

### 2.1.2 Mace Head site

The Mace Head station is located on the west coast of Ireland at 53º20'N, 9 º54'W, 5 m above sea level, and a 23 m-high tower is used to sample the air. Atmospheric concentrations of $CO_2$ have been continuously monitored at Mace Head since 1992 (Biraud et al., 2000; Derwent et al., 2002; Ramonet et al., 2010). Since 2010 a cavity ring-down spectrometer similar to the one used on board the ship (Picarro, G1301), has been used for $CO_2$ and $CH_4$ measurements (instrument owned by the Environment Protection Agency (EPA), Ireland). A second CRDS analyser (Picarro, G2301) was installed in 2013 (instrument owned by Laboratoire des Sciences du Climat et de l'Environnement (LSCE), France), to ensure redundancy of measurements, thus reducing data gaps. Both analysers are calibrated simultaneously every month using a suite of four calibration cylinders whose concentrations span the atmospheric range. Those cylinders have themselves been calibrated at LSCE with WMO/NOAA reference scales (WMO2007 scale for $CO_2$, WMO2004A scale for $CH_4$). In addition to the calibration cylinders, two target cylinders are regularly analysed (short-term target twice a day and long-term target once a month) in order to assess the measurements' repeatability. Over the period 2014-2018 the difference between the assigned values and the values measured every month at Mace Head for the long-term target gas were $0.01 \pm 0.02$ ppm for $CO_2$ (both analysers) and $0.08 \pm 0.19$ ppb and $0.01 \pm 0.17$ ppb for $CH_4$ with analyser G1301 and G2301 respectively. The measurements are processed every day at LSCE (Hazan et al., 2016), ensuring a high level of quality control of the dataset. The maintenance of the analysers is coordinated through close collaboration between LSCE, EPA and the National University of Ireland, Galway (NUIG).

### 2.2 Mass balance budgets

The main underlying assumptions of the mass balance approach used to calculate the spatially-integrated emissions budgets of $CO_2$ and $CH_4$ from the Republic of Ireland, Northern Ireland, Wales and England are five-fold:

• Under westerly wind conditions, the Mace Head station on the West coast of Ireland (53° 20' N, 9° 54' W; 5 m as.l., tower height 23 m) receives relatively clean Atlantic air, whilst the concentrations measured along the ferry route result from enhancement in $CH_4$ and $CO_2$ due to land sources over the travel path of the air mass. The concentrations at Mace Head are representative of the inflow into the British Isles in both space and time.

For each nominal temporal averaging period we assume that:

• The planetary boundary layer (PBL) height is constant over the entire spatial domain bounded to the East and West by the ferry route and the meridian at the Mace Head station location, respectively. The North and South boundaries of the domain are taken as the extrema of the latitudinal range covered by the ferry route.

• The air columns between the land surface and the top of the PBL are well-mixed.

• The horizontal wind direction is uniform.

• There is no mass leakage out of / or ingress into the 3D domain.

In the analysis, conditions are selected to fulfil these assumptions as best as possible. Data screening and quality control are discussed in section 2.2.1, the procedure for estimating background concentrations (baselines) and PBL heights are presented in Sections 2.2.2 and 2.2.3, and, finallty, the mass balance budget and uncertainty calculations are discussed in Sections 2.2.4.

### 2.2.1 Data screening

Prior to time averaging and flux calculation, raw data points were excluded from further processing if any of the following criteria were realised:

• The ship was in port.

• A calibration took place.

- 72-hour back-trajectories (500 m a.g.l) for the Mace Head site and one point along route of the ferry (54.548 °N, 0.233 °W) as calculated with HYSPLIT (NOAA Air Resources Laboratory, 2018) exhibited air flow patterns inconsistent with the mass balance assumptions (i.e. non-westerly flow, evidence of re-circulation).

- The relative wind direction measured on the ship (the fixed reference point being the prow of the vessel) was outside the range 150°-210°. This criterion was used to exclude data points potentially contaminated by on-board activities (e.g. emissions from chimney stacks).

- The wind direction measured on the ship (absolute direction from North, corrected for the movement of the ship) was outside the Westerly range (240° - 300°).

- The wind direction measured at the Mace Head station (data source: Met Éireann, 2018) was outside the Westerly range (240° - 300°).

The temporal coverage of the data points which satisfied the criteria listed above is presented in histogram form in Fig. 4. The full details of the data availability for the study period 2015-2017 are summarised in table S1 of the Supplementary Material.

### 2.2.2 Concentration baselines

The time series of hourly concentrations of $CO_2$ and $CH_4$ measured at Mace Head and filtered for Westerly flow (wind direction range 240° – 300°) were used to construct continuous baselines for the measurement period February 2014 – December 2017 (Fig. 3). The time series of both gases exhibited well-defined seasonal cycles characterised by high concentrations towards the end of the winter and lower concentrations in summer. The baselines were constructed for the data period 01/01/2014 – 31/12/2017 by applying regressions by parts consisting of linear and non-linear (Gaussian) fitting functions over the temporal domain. The composite fitting functions provided smoothing and gap filling of the measured mole fractions time series and were subsequently used to construct continuous time series of background emissions of $CO_2$ and $CH_4$ with a 5-minute time step, which corresponded to the averaging interval used for the data measured on the ferry.

### 2.2.3 Estimation of the planetary boundary layer height

The Weather Research and Forecast model version 3.7.1 (www.wrf-model.org) (Skamarock et al., 2008) was used for this work. The WRF model initial and boundary conditions were derived from the US National Center for Environmental Prediction (NCEP)/National Center for Atmospheric Research (NCAR) Global Forecast System (GFS) at 1.0°×1.0° resolution (National Centers for Environmental Prediction, 2000), including Newtonian nudging every 6 hours. The Yonsei University Scheme (YSU) planetary boundary layer physics option was used here (Hong et al., 2006).

The WRF model domains setup used in this study had three nested domains with horizontal resolution of 0.5°×0.5° for the European domain, 0.16°×0.16° for the British Isles domain, and 0.055°×0.055° for the UK model domain. The vertical column was divided into 21 layers from the surface (bottom layer ~ 50 m) up to 100 hPa (~16 km) in sigma coordinates.

The WRF model hourly output from the UK domain was used to calculate spatial means and standard deviations of the wind speed, wind direction, and the planetary boundary layer height. We estimate the spatial averages at a height of ~450 m (4[th] model layer) for an area defined as follow: lower left corner coordinates of 52.0 latitude and -10.0 longitude and the upper right corner of 57.0 latitude and 3.0 longitude. Time series of hourly averages of wind speed, wind direction and PBL height were constructed for the data period 01/01/2014 to 31/12/2017.

Daily means and standard deviations obtained by averaging the hourly values of the PBL heights derived from WRF for the study period 2015-2017 are presented in Fig. S1 of the Supplementary Material.

### 2.2.4 Mass balance calculations

The flux $F_C$ of species $C$ through a two-dimensional, vertical plane perpendicular to the mean wind direction can be expressed as (Cambaliza et al., 2014; White et al., 1976):

$$F_C = \int_{z_{min}}^{z_{max}} \int_{x_{min}}^{x_{max}} (C - C_b) \cdot U_\perp \, dx \, dz \, , \tag{1}$$

In Eq. (1), $C$ and $C_b$ are the number of moles of species $c$ downwind and upwind of the vertical plane, and $U_\perp$ is the mean wind speed perpendicular to the plane bounded horizontally by $x_{min}$ and $x_{max}$ and vertically by $z_{min}$ and $z_{max}$.

$F_c$ can be expressed explicitly in units of mol.s$^{-1}$ as:

$$F_C = \int_{z_{ground}}^{z_{PBL}} \int_{x_{min}}^{x_{max}} \Delta\chi_C \cdot U \cdot n_{air}(z) \cdot \cos\theta \, dx \, dz \tag{2}$$

Here, $\Delta\chi_c$ is the enhancement of compound $c$ in mol.mol$^{-1}$ above background, $n_{air}(z)$ is the air density at height $z$, $\cos\theta \, dx$ (Fig.

4) is the ship track increment projected onto the crosswind plane and $U$ is the mean wind speed within the PBL obtained by the WRF-model described in Section 2.2.3.

In practice, despite the 3 years of data, no single journey satisfied all the quality control criteria detailed in Section 2.2.1 perfectly for all of the individual 5-minute averaging intervals and we opted to aggregate the good 5-minute data points into 0.2°- wide latitude bins using seasonal grouping for each data year. The baseline mole fractions used to calculate the upwind

enhancement of compound $c$ were time-shifted in order to account for the mean air mass travel time across the domain (time taken to travel West-East from the longitude of the Mace Head station to the location of the ferry at hourly mean wind speed derived from the WRF model; see Table S2 of the Supplementary material for seasonal mean values and standard deviations). Seasonal budgets were then calculated from the aggregated data as:

$$F_C = \sum_{i=1}^{lat \, bins} \underbrace{\Delta x(i)}_{\equiv I} \cdot \underbrace{\overline{\Delta\chi_C(\Delta t) \cdot U(\Delta t) \cdot \cos(\theta(\Delta t)) \cdot \int_{z_{ground}}^{z_{PBL}} n_{air}(\Delta t, z) dz}}_{\equiv II} \tag{3}$$

In Eq. (3), term $I$ is the distance travelled per nominal latitude bin $i$ *along a meridian* (the crosswind projection is done by multiplication with $\cos(\theta)$ in term $II$) and term $II$ is the mean (the horizontal bar denotes averaging), for latitude bin $i$, of the product over all the 5-minute averaging periods ($\Delta t$) that passed the quality control tests.

The total variability on seasonal fluxes was approximated as:

$$\Delta F_c = \sqrt{\sum_{i=1}^{lat \, bins} \Delta x(i) \cdot \sigma^2 \left( \Delta\chi_C(\Delta t) \cdot U(\Delta t) \cdot \cos\theta(\Delta t) \cdot \int_{z_{ground}}^{z_{PBL}} n_{air}(\Delta t, z) dz \right)} \tag{4}$$

where $\sigma$ denotes the standard deviation of the mean. Finally, the annual budgets were obtained by summing the seasonal budgets.

Uncertainty and error propagation

In addition to the temporal variability $\Delta F_c$, (Eq. 4) we calculated the uncertainty on the total fluxes arising from the uncertainties on the individual terms of the mass balance equation. Noting that $dx$ represents the distance travelled by the ship with speed

$v_{ship}$ during the infinitesimal time interval $dt$, Eq. 2 can be reformulated to express the partial flux $f_c$ through a 2-dimensional plane spanning the horizontal distance $dx$ as a function of $v_{ship}$ and $dt$ (Eq. 5).

$$f_C = \int_{z_{ground}}^{z_{PBL}} \Delta\chi_C \cdot U \cdot n_{air}(z) \cdot \cos\theta \cdot v_{ship} \, dt \, dz \tag{5}$$

Applying the rules of error propagation, the error on the flux term $f_c$ ($\delta f_c$) is given by (with $N_{air}$, the value of the integral of $n_{air}(z)$ evaluated over time step $dt$):

$$\frac{\delta f_c}{|f_c|} = \sqrt{\left(\frac{\delta\chi_c}{\chi_c}\right)^2 + \left(\frac{\delta U}{U}\right)^2 + \left(\frac{\delta\cos\theta}{\cos\theta}\right)^2 + \left(\frac{\delta v_{ship}}{v_{ship}}\right)^2 + \left(\frac{\delta dt}{dt}\right)^2 + \left(\frac{\delta\int_{z_{ground}}^{z_{PBL}} n_{air}(z) \, dz}{N_{air}}\right)^2} \tag{6}$$

Assuming that, (a) the uncertainty on $dt$ is negligible, and (b) the uncertainty on the PBL height ($z_{PBL}$) is the dominant error term in the integral of $n_{air}(z)$ between height $z_{ground}$ and $z_{PBL}$, Eq. 6 can be approximated as:

$$\frac{\delta f_c}{|f_c|} \approx \sqrt{\left(\frac{\delta \chi_c}{\chi_c}\right)^2 + \left(\frac{\delta U}{U}\right)^2 + \left(\frac{\delta cos\theta}{cos\theta}\right)^2 + \left(\frac{\delta v_{ship}}{v_{ship}}\right)^2 + \left[\frac{\left(n_{air}(z_{PBL}) - n_{air}(z_{ground})\right).\delta z_{PBL}}{N_{air}}\right]^2} \tag{7}$$

Finally, similarly to Eq. 4, the total error on the flux $F_c$ ($\delta F_c$) calculated for a complete transect of the ship between $x_{min}$ and $x_{max}$ is given by:

$$\frac{\delta F_c}{|F_c|} = \sqrt{\Sigma_i^N \left\{\left(\frac{\delta f_c}{|f_c|}\right)_i\right\}^2} \tag{8}$$

The standard deviations of the individual terms in Eq. 7, calculated for each 5-minute averaging period and averaged over each nominal latitude bin, were used as proxies for uncertainties. Table S2 of the Supplementary Material summarises the total uncertainty on the calculated emissions budgets and the relative contributions of the individual terms in Eq. 7.

## 3. Results

### 3.1 Seasonal and annual fluxes

The fluxes of $CH_4$ (Fig. 5) and $CO_2$ (Fig. 6) calculated from measurements on board the North Sea ferry were variable in space (over the latitude range 51.35° – 56.15° N) and time. The calculated emissions of $CO_2$ and $CH_4$ had maxima in winter (DJF; 379.1 ± 26.2 Tg $CO_2$, 0.89 ± 0.08 Tg $CH_4$; 2016 & 2017 winter data only). Emissions minima were observed in summer (JJA; 123.6 ± 76.9 Tg $CO_2$; 0.38 ± 0.25 Tg $CH_4$; Table 3). Springtime (MAM) emissions were 161.5 ± 30.9 Tg for $CO_2$ and 0.55 ± 0.08 Tg for $CH_4$ and, in autumn (SON), the measured emissions were 250.2 ± 200.1 Tg for $CO_2$ and 0.72 ± 0.40 Tg for $CH_4$. For $CO_2$ and $CH_4$, a statistically significant difference in seasonal budgets was found between winter and spring as well as between winter and summer.. For both gases, the differences in emissions between spring and summer were not statistically significant, whilst, for autumn, the total uncertainty was large (80% uncertainty for $CO_2$ and 56% for $CH_4$). Annual budgets, estimated from seasonal values, were 914.4 ± 218.1 Tg for $CO_2$ and 2.55 ± 0.48 for $CH_4$. Without accounting for seasonality (i.e. using all data without seasonal segregation, which could weight towards the periods of the year for which the most data were available), the emissions budgets were 708.3 ± 270.4 Tg for $CO_2$ and 2.1 ± 0.67 for $CH_4$.

In winter, spring and autumn, the largest fluxes of both gases were found in a broad central latitudinal band (52 °N – 54 °N; Fig. 5 and Fig. 6). The lowest emissions were observed in summer across the entire spatial domain and they exhibited a smaller increment in the 52 °N – 54 °N band compared to the fringes of the domain than in other seasons.

### 3.2 Diurnal variability

There were differences between day (defined arbitrarily as 09:00 to 18:00) and night fluxes, particularly in spring and summer (Fig. 7, Fig. 8). Median daytime $CO_2$ fluxes were negative for latitudes in the range 54.5 °N to 55.9 °N in spring; in summer, negative $CO_2$ fluxes were found at 54.5 °N and 55.3 – 55.5 °N. It is important to note that air mass transit time between the in- and out-flow points of the domain varied from a median of 11 hours in winter to 19 hours in summer, which means that the day and night periods did overlap.

Seasonal and annual budgets were re-calculated using day and night fluxes weighted by day length (Table 3) in order to assess the impacts of uneven day/night data density distributions over the spatial domain caused by the relatively slow travel speed of the ship and the random data gaps introduced by changing wind direction and measurement downtime. The annual budgets calculated with day/night flux segregation were smaller than those obtained without day/night partitioning but the differences were not statistically significant and, in general, separating fluxes into day and night components increased the uncertainties on the final budgets both at the seasonal and annual levels.

The annual budgets calculated using all available data were smaller than those obtained from seasonal budgets (both with and without day/night segregation), however, the only statistically significant difference was between the annual budget of $CO_2$ obtained from seasonal data and the budget estimated with day/night weighting but without seasonality.

The annual budgets for both gases obtained without accounting for the seasonality in data coverage were consistent with inventory data but the measurement uncertainties were large (36% and 32% for $CH_4$ with and without day/night weighting, with counterpart uncertainties on $CO_2$ budgets 42% and 38%, respectively). The annual budgets of $CH_4$ obtained from seasonal budgets were in good agreement with inventory data, with uncertainties of 12% and 19% for estimates calculated with and without day/night weighting, respectively. In contrast, $CO_2$ budgets were almost double the inventory value, with uncertainties of 14% and 24% for estimates calculated with and without day/night weighting.

The seasonal mass balance fluxes of $CH_4$ and $CO_2$ calculated from concentration measurements on the ferry were compared to known land sources and meteorological drivers of these gases. For both gases, there was a strong, positive correlation between seasonal emissions measured on the ferry and consumption of natural gas in the UK (Fig. 9). The correlation between GHG emissions and mean air temperature was negative and statistically significant (Fig. 10).

## 4. Discussion

The mass balance approach presented here relies on simplifying assumptions to derive GHG budgets for a large part of the British Isles. The main assumptions are that a) the air masses travel West to East, b) the PBL height is constant over the spatial domain for each nominal averaging period, c) there is no loss or input of mass into the domain other than from land sinks/sources, and d) the air is well-mixed over the entire PBL height.

The data were filtered for westerly flow based on air mass back trajectories obtained from the HYSPLIT Trajectory model (NOAA Air Resources Laboratory, 2018) daily 72-hour runs at two coordinates: the Mace Head reference site and one ferry position halfway along its route. The back trajectories were run daily, commencing at midnight, and the air mass histories were assumed to be valid for an entire 24-hour period and for the entire spatial domain. Of the four main assumptions listed above, points c) and d) are the most subjective because they could not be verified nor quantified. Assumption a) (air mass travel from West to East) can be considered to be reasonably well-constrained owing to the data screening procedure at the pre-processing stage. Violations of the stationarity assumption (point b) due to significant changes in the mean PBL height at sub-hourly time step would either be captured, in part or entirely, during the next hourly averaging period, or go unnoticed in the case of very transient non-stationary events. Whilst the temporal variability of the mean PBL height for the spatial domain considered can be quantified and propagated through the emissions budgets calculations as measurement uncertainty, the potential bias between model output and observations is unknown. Recent studies have compared different WRF parametrisation schemes with observed PBL height and found that, in general, the YSU scheme used in this study performs reasonably well in terms of predicting PBL height with minimum bias typically observed before midday (Hu et al., 2010, Banks et al., 2016, Tyagi et al., 2018, Xu et al., 2018); however these studies also highlighted that model performance can vary significantly between sites and time of day, and that YSU tends to underestimate the PBL height over the sea (Tyagi et al., 2018). Comparisons between observations and model outputs of wind speed profiles for different parametrisation schemes also found substantial variability, both intra- and inter-model, with the YSU scheme exhibiting a tendency to overestimate wind speeds (Balzarini, 2014, Tyagi, 2018). The formation of sea breezes adds another level of complexity to the modelling of PBL height and wind speed, in particular in the southern North Sea where the orientation of the coastlines and their proximity to one another have been shown to induce sea breeze formation and to influence sea breeze type and offshore extent (Steele et al, 2013; Steele et al., 2015). Furthermore, not all WRF parametrisation schemes are equal in performance with respect to sea breeze conditions; recent

studies show that the YSU scheme used here exhibited the smallest bias for wind speeds measured onshore under complex sea breeze conditions (Steele et al., 2015) and that it also captured the temporal evolution of the atmospheric boundary layer height better than other schemes (Salvador et al, 2016).

Intrinsic, unquantifiable biases on the mixing layer heights and mean wind speeds derived from the WRF model are hence likely. Wind speed and enhancement above background concentration were found to be to dominant uncertainty terms, jointly accounting for over 80% of the total uncertainty in all seasons (Table S2 of the Supplementary Material). In contrast, nudging the baseline concentrations measured at Mace Head by a time lag estimated from the mean air mass travel time had only a very modest impact on the final budgets (Table S2). The two measures of errors proposed in this paper (based on temporal variability and total uncertainty through error propagation) yield on the whole comparable results, with the main discrepancy found for the autumn budget (years used: 2015-2017) where the total uncertainty was almost four-fold the value obtained by considering the temporal variability alone. The autumn uncertainty was brought in line with the temporal variability estimate for both gases when the day/night weighting was applied. Whilst the variability and the total uncertainty are useful as first approximations for the confidence in the emission budgets, they should be treated as potential lower limits because of the unquantified bias between WRF model outputs and actual values of the PBL height and wind speed.

The fluxes calculated under this data filtering regime were assumed to be representative of surface emissions and uptake over the land masses bounded by the spatial domain and local influences (due to e.g. localised air re-circulation) were assumed to be negligible. This assumption could not be tested on a point-per-point basis but the latitudinal trends for both $CH_4$ and $CO_2$ at the seasonal time scale (Fig. 5 and Fig. 6) are consistent with the demographics and the known spatial distributions of sources of GHGs over the latitudinal range considered. In particular, the emission peaks for $CO_2$ observed around 52.5 °N and 54 °N coincide with major urbans centres in the British Midlands, namely Birmingham and Manchester/Liverpool/Leeds/Sheffield, and Dublin further upwind in the Republic of Ireland. These conurbations are reported to be significant sources of $CO_2$ by the UK's official National Atmospheric Emissions Inventory (NAEI, 2018a). The calculated $CH_4$ emissions were elevated in the 52.5 °N - 54 °N latitude band compared to the fringes of the domain. This agrees with the NAEI UK $CH_4$ map (NAEI, 2018a), which shows large emissions from the western parts of England in that latitude band. The NAEI reports substantial $CH_4$ emissions from the Cornwall area (SW England; latitudes < 51.3 °N), which might not always have registered in their entirety by the measurement system on the ferry because the port of Zeebrugge – the starting/end point of the vessel's route – lies at 51.21 °N.

The negative daytime fluxes of $CO_2$ registered in summer for latitudes > 54.5 °N are consistent with the demographics, topography and land-use of the northern parts of England and of Northern Ireland; these areas are less populated than the southern parts, host the hills and mountains of the Lake District and the Northern Pennines and the land-use consists largely of grasslands. The combination of these factors (lower density of anthropogenic sources and higher density of biogenic sinks compared to southern parts of the UK) can explain the net negative fluxes of $CO_2$ measured during the daytime in spring and summer. Whilst the observed lower emissions of $CO_2$ in the northern parts of the spatial domain are consistent with the spatial distribution of emissions from NAEI data (NAEI, 2018a), Polson et al. (Polson et al., 2011) reported substantial summertime emissions from Ireland and Northern Ireland which should cancel out the sink terms in Northern England when integrating along a latitude bin. The fact that negative and very low summer emissions were derived by the ferry mass balance approach could indicate that measurements on board the ferry were more sensitive to sources and sinks in the eastern parts of the domain sampled because of, a) violation of the simplifying assumption that there is no loss of mass out of the domain, b) imperfect vertical mixing or, c) local air circulation which would not have been resolved by the HYSPLIT air mass histories. Alternatively, the mass balance estimates are real and the high $CO_2$ emissions assigned to Ireland in the aircraft inversion

model are measurement artefacts caused by the venting of the nocturnal boundary layer as postulated by Polson et al. (Polson et al., 2011).

There was no statistically significant difference between day and night fluxes for $CH_4$, which could be because, a) the major sources of this gas in the British Isles (livestock - enteric fermentation and manure management - and waste treatment related
emissions – landfills and waste water - 52.8% and 39.1% of the total $CH_4$ budget for the UK, respectively; BEIS, 2017) do not have marked diurnal cycles, b) the mass balance approach could not resolve them, c) the transit time of the air masses over the spatial domain blurred the potential differences between day and night time emissions or, d) the $CH_4$ signal measured on the ferry was contaminated and did not reflect emissions from the land surface. Due to the temporal and spatial averaging carried out to derive emission estimates from the ferry measurements, and due to the diffuse spatial distribution of the dominant
land sources of $CH_4$, it seems likely that relatively small diurnal variations (e.g. studies indicate diurnal cycles in $CH_4$ emissions from dairy farms - (VanderZaag et al., 2014) - and from landfills sites - (Borjesson and Svensson, 1997)) would not be resolved by the mass balance approach.

At the seasonal time scale, the fluxes of $CH_4$ and $CO_2$ were both strongly correlated with UK natural gas usage (Fig. 9; BEIS, 2018); this provides confidence that the fluxes calculated by the mass balance approach can be related to physical emissions
within the spatial domain and that the data filtering and quality control criteria excluded data points potentially contaminated by emissions from the ship. The statistically significant linear correlations between derived $CH_4$ and $CO_2$ fluxes and natural gas usage do not demonstrate causality, but suggest that the sources of these two GHGs within the domain sampled have seasonal dynamics similar to those of natural gas usage. However, both $CH_4$ and $CO_2$ emissions exhibited a weaker correlation with mean seasonal air temperature than with natural gas usage (Fig. 10), and this may indicate that natural gas consumption
is a causal driver rather than a proxy for another underlying variable. Whilst it is reasonable to infer that both $CO_2$ and $CH_4$ would increase in line with an increasing demand for natural gas during the colder months, the NAEI (NAEI, 2018b) attributes only 15% of annual $CH_4$ emissions to fuel-related sources (combustion and fugitive emissions); this does not tally with the ~ 100% increase in $CH_4$ emissions between winter and summer which is accompanied by a similar increase in natural gas usage. An unexpectedly large diurnal and seasonal variability in the $CH_4$ flux was observed from direct flux measurements above
London (Helfter et al., 2016) and this suggested that pressure variations in the gas supply network in respond to gas demand may have a significant impact on urban emissions. Fugitive emissions from the network may be underestimated in the NAEI.

Seasonality in methane emissions from landfills has also been reported, with higher emissions sometimes observed in winter and autumn (Borjesson and Svensson, 1997; Chanton and Liptay, 2000). The explanation for this is that net $CH_4$ emissions from landfill emissions can be largely regulated by methane oxidation in the top layer of the landfill cover soil: oxidation is
limited by soil temperature and the methane-oxidising potential decreases in autumn and winter because of lower soil temperatures, which results in an increase in methane emissions during the colder seasons. Riddick et al. (Riddick et al., 2017) reported a 71% winter-to-summer reduction in $CH_4$ emissions from a waste treatment park near Haddenham, England. Central England has the largest densities of waste treatment and landfill sites, which might explain the statistically significant, linear anti-correlation between seasonal $CH_4$ emissions and mean air temperature found in this region (Fig. 11). This is a remarkable
result, which demonstrates the merit of this simple mass balance approach. In the other two regions considered ("N" and "S", i.e. north and south of the central region denoted as "MID"), there was no compelling correlation between $CH_4$ emissions and mean air temperature. This suggests that the dominant sources of this GHG in the N and S regions differ from the ones in the central region.

For $CO_2$, the seasonal emissions had statistically significant correlations to mean air temperature in the central and northern
regions whilst the linear correlation was only marginally non-significant in the southern region. This is consistent with, a) the

seasonality of natural gas usage (the NAEI attributes ~ 50% of annual $CO_2$ emissions to fuel combustion processes such as domestic and industrial gas usage; NAEI, 2018b) and, b) the seasonality of $CO_2$ uptake by vegetation.

Contrary to our findings, the UK $CH_4$ emissions derived by inverse modelling using concentration data from four tall tower sites distributed across the UK and Ireland did not exhibit any clear seasonality over the period August 2012 – August 2014 (Ganesan et al., 2015), but the range of emissions (1.65 Tg to 2.67 Tg) was consistent with the ferry measurements ($1.52 \pm 1.0$ Tg to $3.56 \pm 0.32$ Tg).

For $CH_4$, all four annual budgets calculated using all the available data for the 2015-2017 period were consistent with the inventory values for the UK (excluding Scotland) and the Republic of Ireland, as well as with top-down modelling estimates (Table 3). Temporal data aggregation (i.e. not considering seasonality) increased the uncertainty on the final budget (36% and 32% uncertainty for annual budgets derived with and without considering differences in day and night emissions, compared to 12 % and 19% for the budgets where seasonality was factored in) and it seems therefore that this approach should be discarded. The difference between the annual $CH_4$ budgets calculated with and without day/night segregation but with seasonality was within the uncertainty of the individual estimates and since we found no compelling evidence of diurnal trends, we arrive at $2.55 \pm 0.48$ Tg $y^{-1}$ as our final estimate of the methane emissions from the UK (excluding Scotland) and the Republic of Ireland for the period 2015-2017.

Following the same argument regarding temporal data aggregation, we derive an annual emission budget for $CO_2$ of $881.0 \pm 125.8$ Tg $y^{-1}$, which is the estimate obtained from seasonal budgets with day/night segregation because we found indications of diurnal trends in some parts of the spatial domain.

This value is over two-fold the inventory estimate of 422.7 Tg, but contrarily to $CH_4$, $CO_2$ has significant biogenic sources (e.g. the $CO_2$ exhaled by the 65 million-strong human population within the spatial domain considered is of the order of 18 Tg $y^{-1}$ (Moriwaki and Kanda, 2004)) and sinks (vegetation uptake) which are not accounted for by the anthropogenic atmospheric emissions inventories; a direct comparison with the inventory is hence not possible. Polson et al. (2011) derived an annual budget for $CO_2$ of $620 \pm 105$ Tg $y^{-1}$ from a series of flights around Britain in the summer of 2005 and September 2006. Using only summer data, in order to emulate the temporal upscaling done by Polson, we arrive at an annual $CO_2$ budget of $511 \pm 308$ Tg $y^{-1}$, which agrees with the 2011 aircraft study within measurement uncertainty. Whilst the seasonality of $CO_2$ emissions cannot be disregarded, comparing our summer time budgets with the aircraft study provides an independent validation of the ferry mass balance approach and gives us confidence in the method despite the simplifying assumptions that underpin it. Finally, we compared the ferry-derived summertime estimates for the southern region, filtered with a narrow 260°-280° wind direction window, to the fluxes of $CO_2$ and $CH_4$ obtained in 2012 by airborne measurements in the greater London area (O'Shea et al., 2014). The ferry fluxes of both gases ($CH_4$: $0.049 \pm 0.020$ Tg.season$^{-1}$; $CO_2$: $24 \pm 15$ Tg.season$^{-1}$) compared reasonably well with the ones from the airborne campaign ($CH_4$: $0.034 \pm 0.002$ Tg.season$^{-1}$; $CO_2$: $13.4 \pm 1.2$ Tg.season$^{-1}$), but clearly also include sources upwind and downwind of the greater London area. The uncertainty was large for both gases, which is unsurprising considering the length of the averaging period (summers of 2015 and 2016), but this comparison with another independent measurement further consolidates the confidence in the method and in the overall annual budgets for $CH_4$ and $CO_2$.

## 5. Conclusions

Applying a mass balance approach to continuous measurements of $CO_2$ and $CH_4$ in the outflow, using a ship of opportunity, we estimated the net annual emissions of $CH_4$ from the UK (excluding Scotland) and the Republic of Ireland, averaged over

the 2015-2017 period, to be $2.55 \pm 0.48$ Tg, which is consistent with the combined 2.29 Tg reported to the United Nations Framework Convention on Climate Change. The annual $CO_2$ budget obtained by mass balance ($881.0 \pm 125.8$ Tg) was over two-fold the inventory value (422.7 Tg), but a direct comparison is not possible for this gas because the atmospheric inventory only accounts for anthropogenic sources (BEIS, 2016). Instead we compared our $CO_2$ budget estimate with previous airborne studies, one for the UK as a whole and the second one for the greater London area and found good agreement with both. The mass balance approach presented here does not provide direct source apportionment information, but the latitudinal emissions patterns observed for both $CH_4$ and $CO_2$ were generally consistent with known spatial distributions of sources and sinks. Assuming that the atmospheric emissions inventory captures all anthropogenic emissions, we estimate that the net biogenic component of the measured $CO_2$ annual budget was 458.7 Tg, which corresponds to 52% of the total emissions. We detected marked seasonality in the emissions of both gases with lower values in the summer, and the seasonal budgets had statistically significant correlations with natural gas and mean air temperature. We attribute the two-thirds decrease in $CO_2$ emissions between winter and summer for $CO_2$ to the superposition of the reduction in demand for fossil fuels and an increase in the biogenic sink during the summer. For $CH_4$, we attribute the seasonal variability of the measured fluxes to natural gas consumption and to the waste management sector where temperature has been shown to control the methane oxidising potential of landfill cover soil and, thereby, the net emissions. With this study, we validated the atmospheric emissions inventory of $CH_4$ for the UK (excl. Scotland) and Ireland, quantified the biogenic component of the annual $CO_2$ budget and derived seasonal emissions budgets for both gases. Finally, we demonstrated that $CH_4$ emissions are strongly seasonal even at such a relatively large spatial scale, which highlights the importance of taking meteorological drivers such as air temperature into account in future "bottom-up" budgets.

**Author contribution**

Carole Helfter led the writing of the manuscript with contributions from all co-authors.

**Acknowledgments**

The GAUGE project was funded by the UK Natural Environment Research Council under grant reference NE/K002449/1. We are grateful to DFDS Seaways for authorising the research activities on board the Rosyth–Zeebrugge commercial ferry and we thank the captains and crews of the Longstone (now the Finnmerchant) and Finlandia Seaways for access to the ships and for assistance with the day-to-day research operations. We also acknowledge the contributions of Gerry Spain (NUIG), Victor Kazan (LSCE) and Damien Martin (EPA, Ireland) for the maintenance of the analysers at the Mace Head measurement station.

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

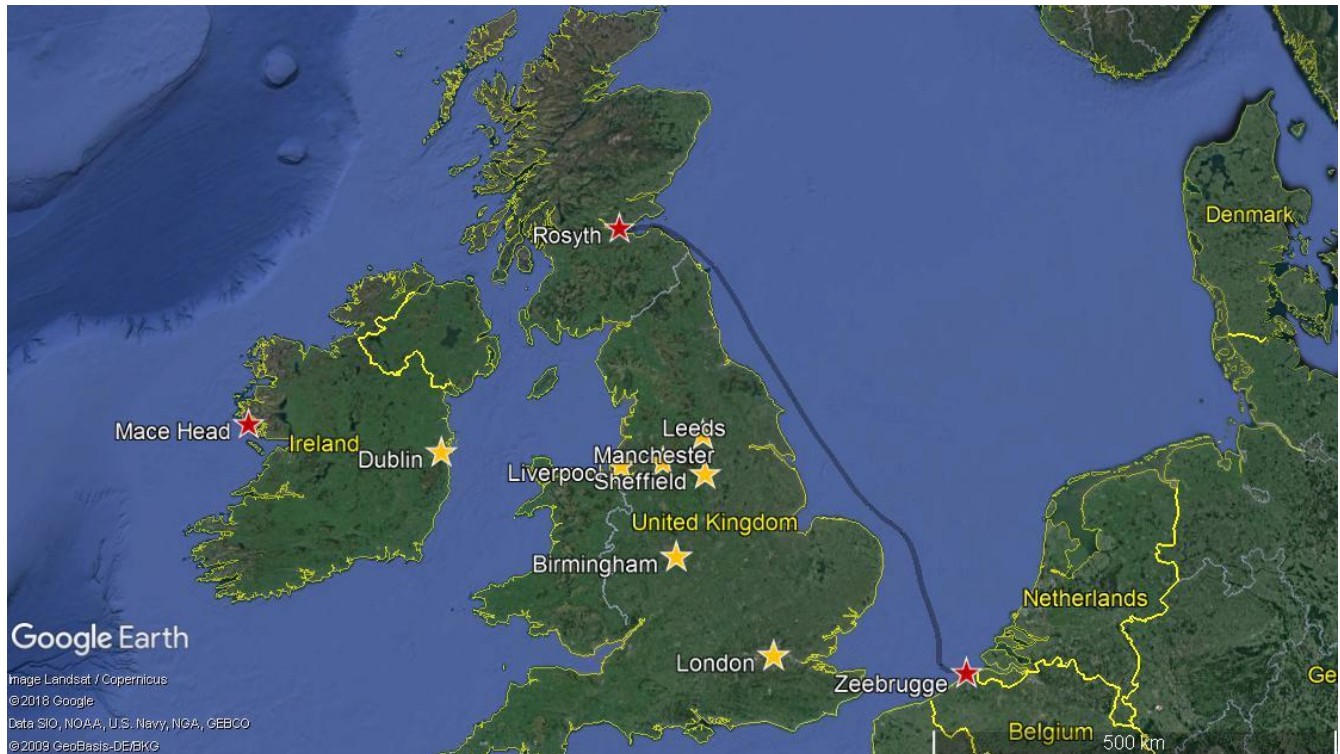

**Figure 1.** Google Earth map centred on the United Kingdom and Ireland. The route of the ferry is indicated by a dark blue line joining the ports of Rosyth (Scotland, UK) and Zeebrugge (Belgium). The location of the Mace Head measurement station on the west coast of Ireland, which provided the carbon dioxide and methane concentration baselines, is indicated by a red star. The cities indicated by yellow stars are

5 locations of interest cited in the discussion (Section 4).

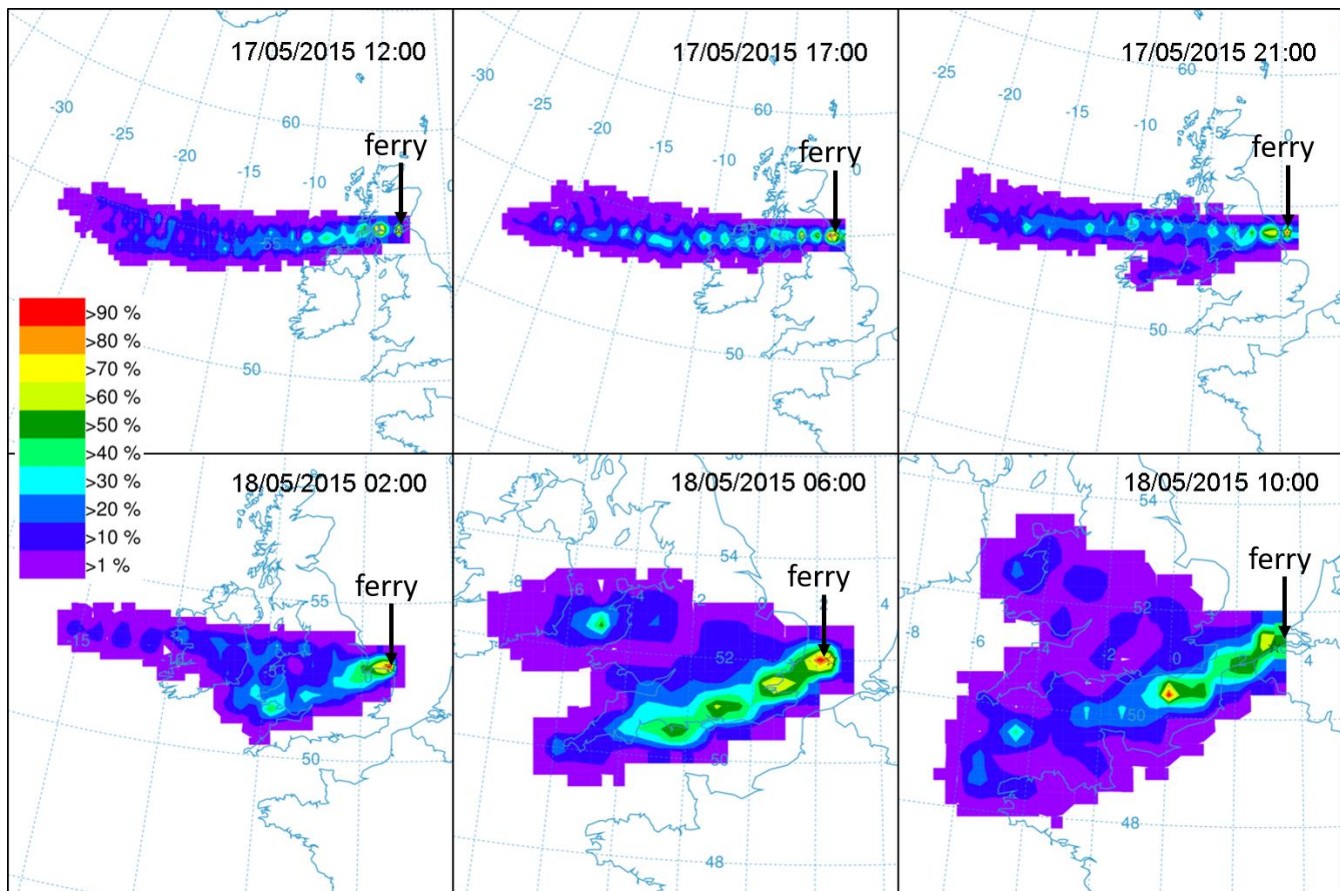

**Figure 2.** Backward trajectory frequencies for a South-bound sailing with westerly wind conditions (sailing start 17/05/2015 12:00, end 18/05/2015 10:00). The coloured contours represent the normalised frequency counts (number of end points in a 0.5°x 0.5° grid cell divided by the maximum number of end points in any grid cell, expressed as a percentage) and the source corresponds to the location of the ferry (indicated by an arrow). The trajectories were run backward for 24 hours at 3-hour intervals using GDAS 1-degree global meteorology (NOAA, 2018).

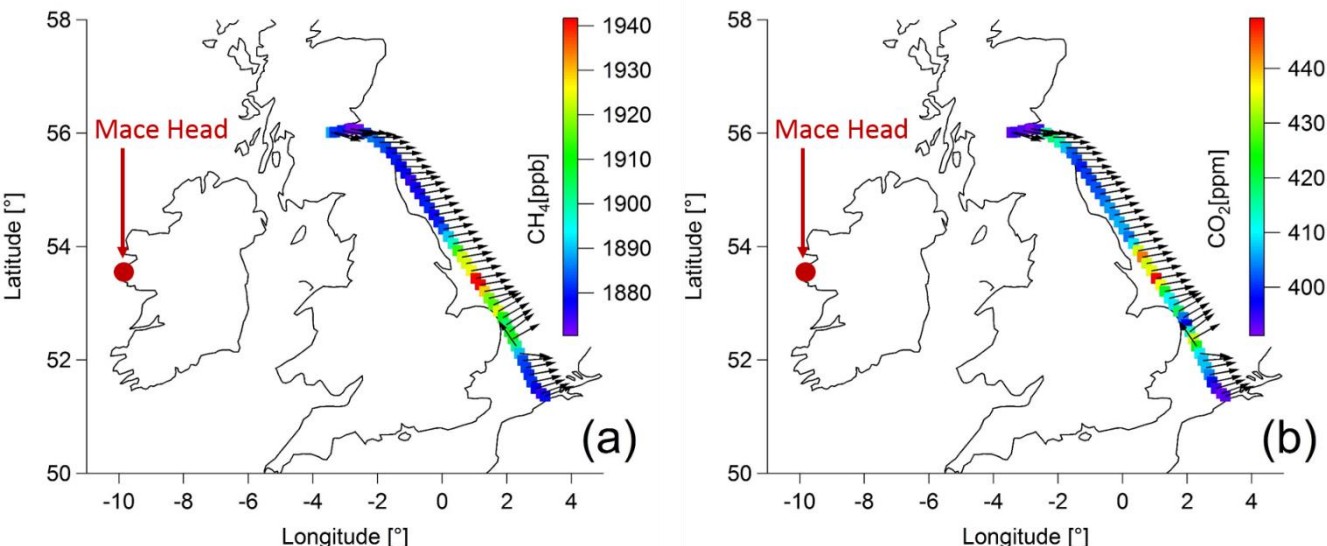

**Figure 3.** Half-hourly averages of (a) CH$_4$ and (b) CO$_2$ mole fractions measured on board the freight ferry during the South-bound journey on 29-30 July 2014. The arrows represent wind direction.

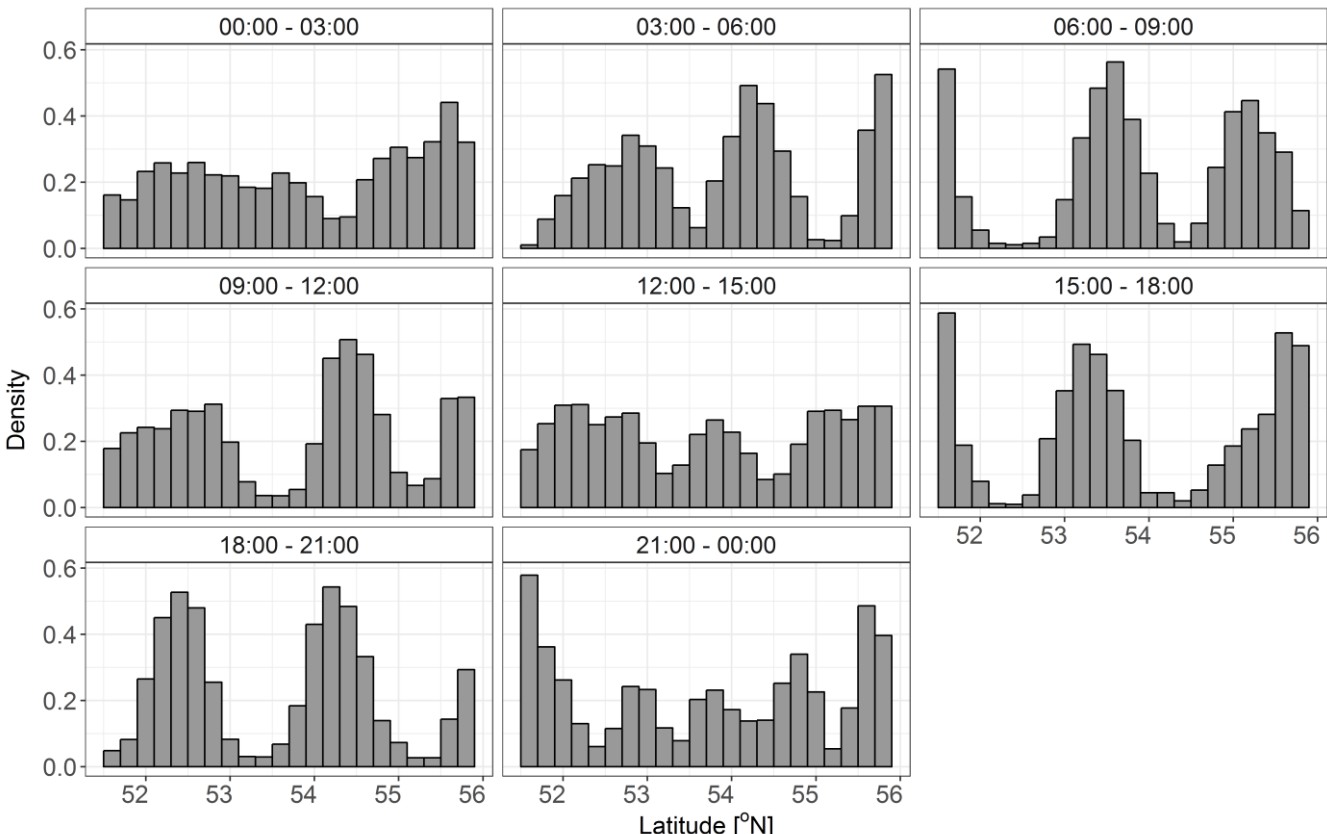

**Figure 4.** Temporal coverage of the latitudinal range (0.2° bins) spanned by the ferry route expressed as a counts density (frequency of occurrence normalised by the total number of observations in each latitude bin) for all data points which satisfied the data screening criteria (section 2.2.1) during the measurement period 01/01/2015 – 31/12/2017.

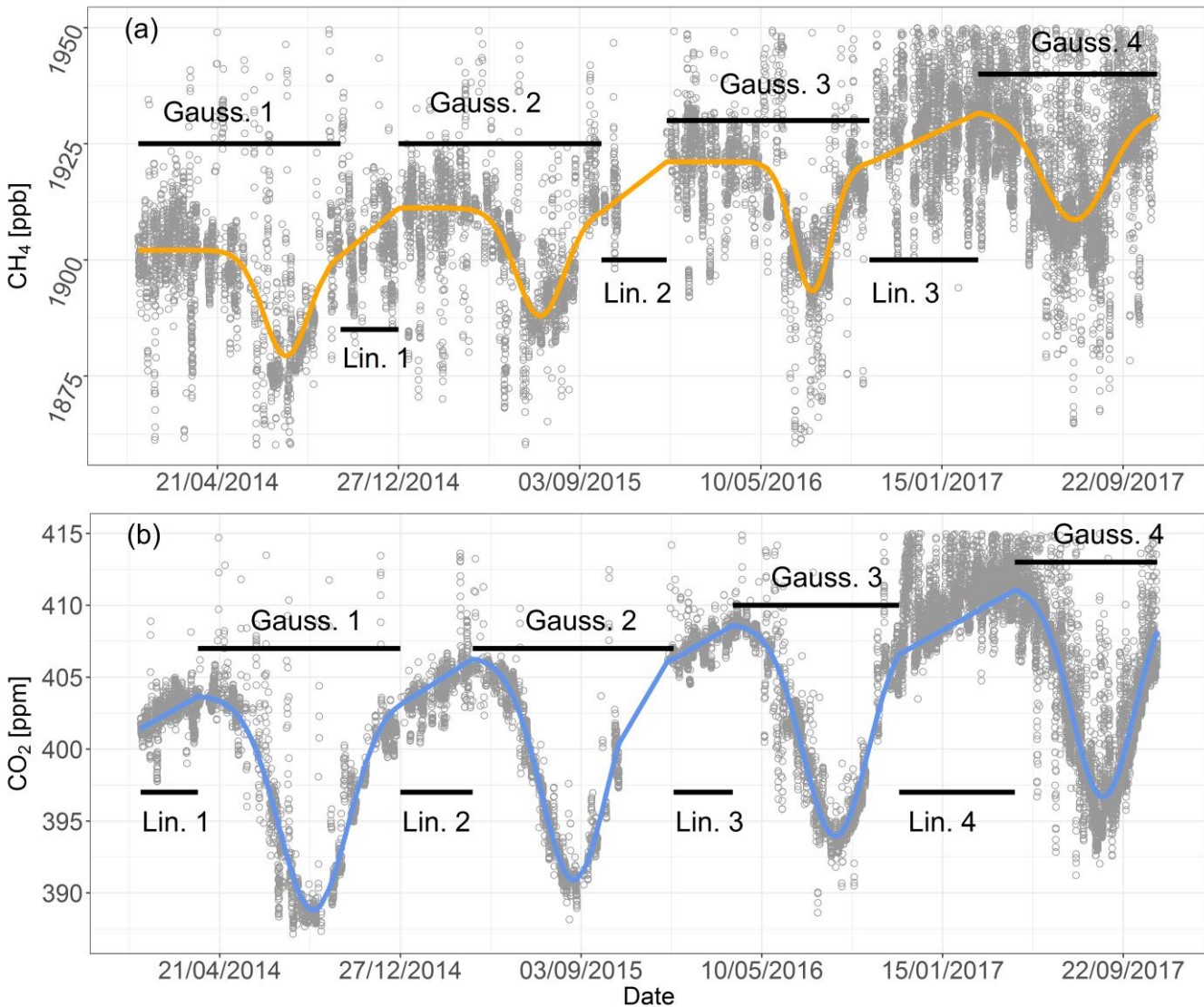

**Figure 5.** Hourly time series filtered for Westerly wind directions (range 150° – 210°) measured at the Mace Head station (open circles) of (a) CH$_4$ and (b) CO$_2$ mole fractions. Smoothing and gap filling of the original time series was achieved by applying linear (Lin.) and non-linear (Gauss.) regressions by parts for the data period 01/01/2014 – 31/12/2017 (solid lines).

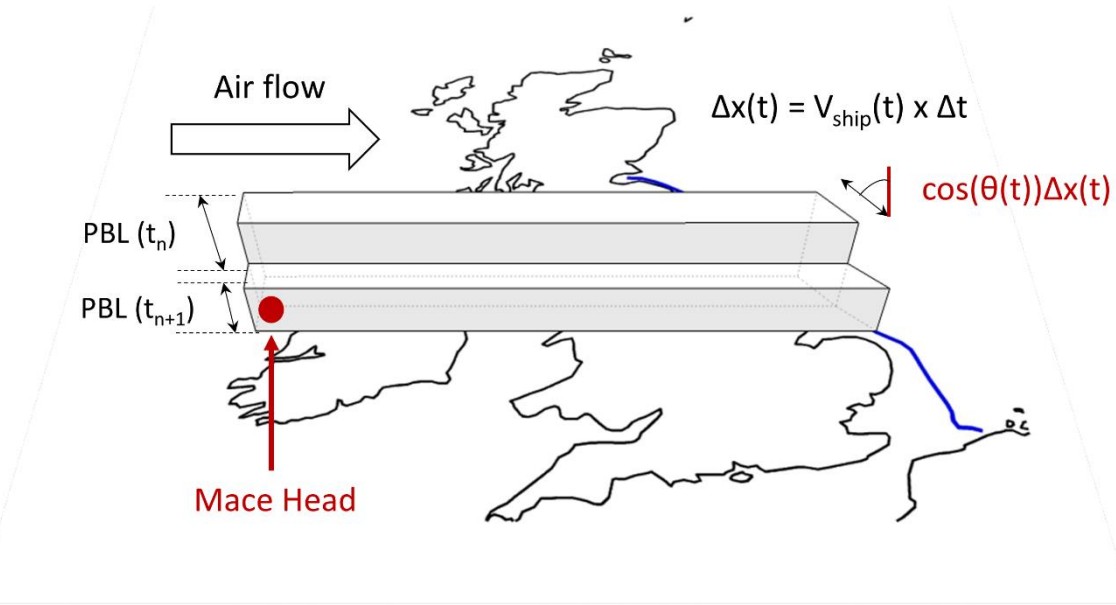

**Figure 6.** Schematic of the mass balance flux calculation procedure expressed in Eq. (2). The total flux is the sum of flux elements through a vertical surface of height that of the planetary boundary layer (PBL) height and width the ship track increment ($\Delta x = v_{ship}(t) \, \Delta t$) during a nominal averaging time interval $\Delta t$ projected onto the cross-wind direction ($\cos((\theta(t)) \, \Delta x$).

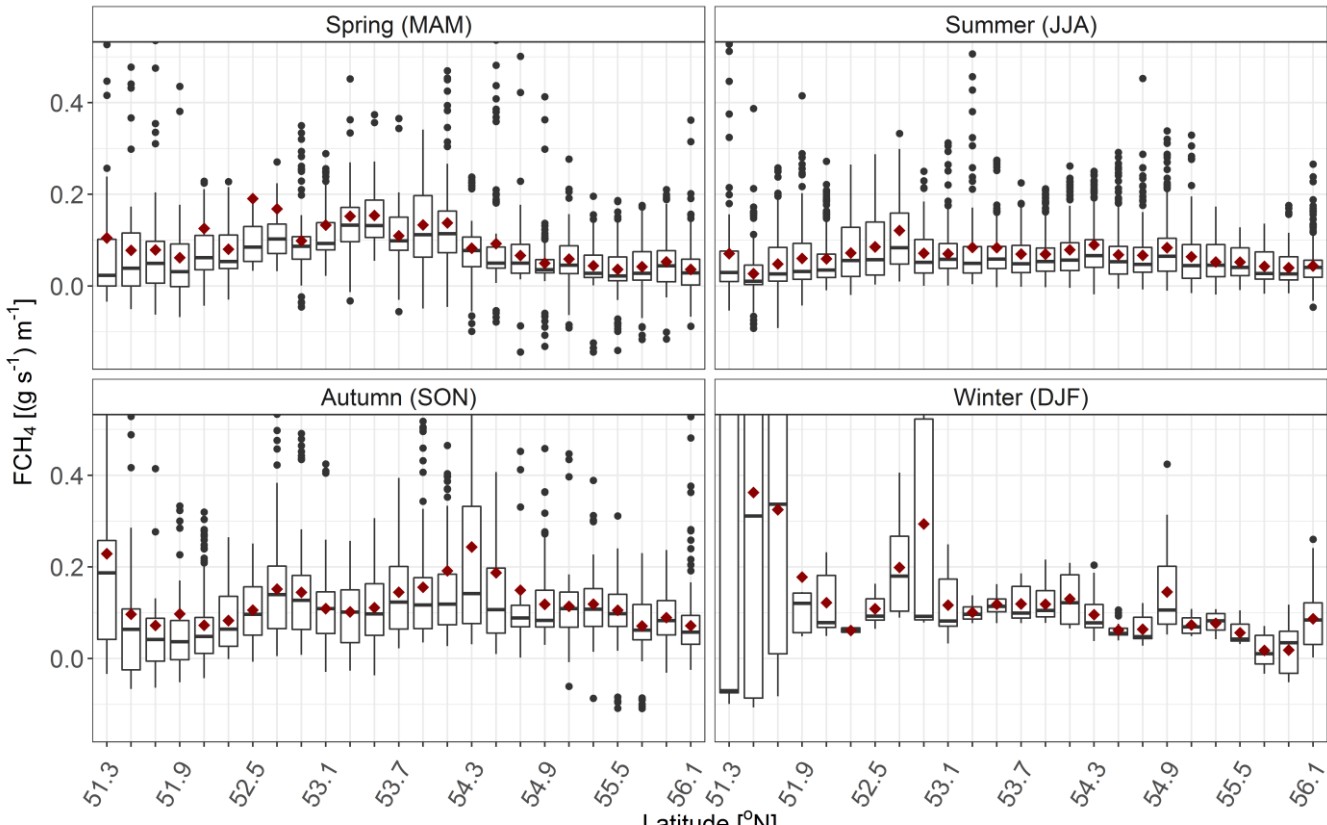

**Figure 7.** Box and whisker plots of 5-minute binned averages of $CH_4$ fluxes along the route of the ferry (latitude bin width: 0.2°). The horizontal bar within each box corresponds to the median for a given latitude bin, the upper and lower hinges represent the 75th and 25th quantiles, respectively, and the upper and lower whiskers indicate the largest/smallest observation less/greater than or equal to upper/lower hinge +/- 1.5 * IQR (Inter-Quantile Range), respectively. The outliers are represented by solid circles and arithmetic means by red diamonds. The flux is integrated over the height of the planetary boundary layer and expressed in units of mass flux per meter travelled crosswind within each latitude bin per unit time.

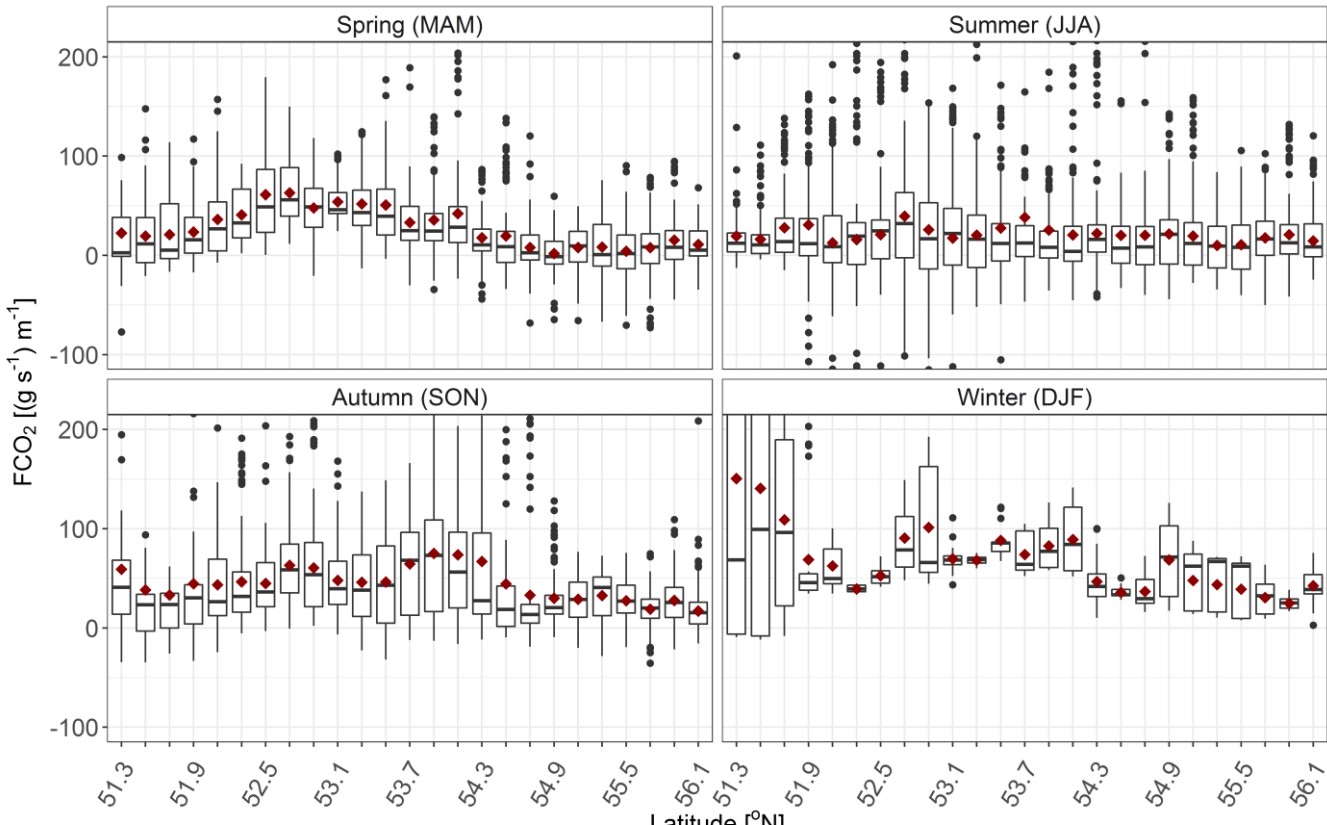

**Figure 8.** Box and whisker plots of 5-minute binned averages of $CO_2$ fluxes along the route of the ferry (latitude bin width: 0.2°). The horizontal bar within each box corresponds to the median for a given latitude bin, the upper and lower hinges represent the 75th and 25th quantiles, respectively, and the upper and lower whiskers indicate the largest/smallest observation less/greater than or equal to upper/lower hinge +/- 1.5 * IQR (Inter-Quantile Range), respectively. The outliers are represented by solid circles and arithmetic means by red diamonds. The flux is integrated over the height of the planetary boundary layer and expressed in units of mass flux per meter travelled crosswind within each latitude bin per unit time.

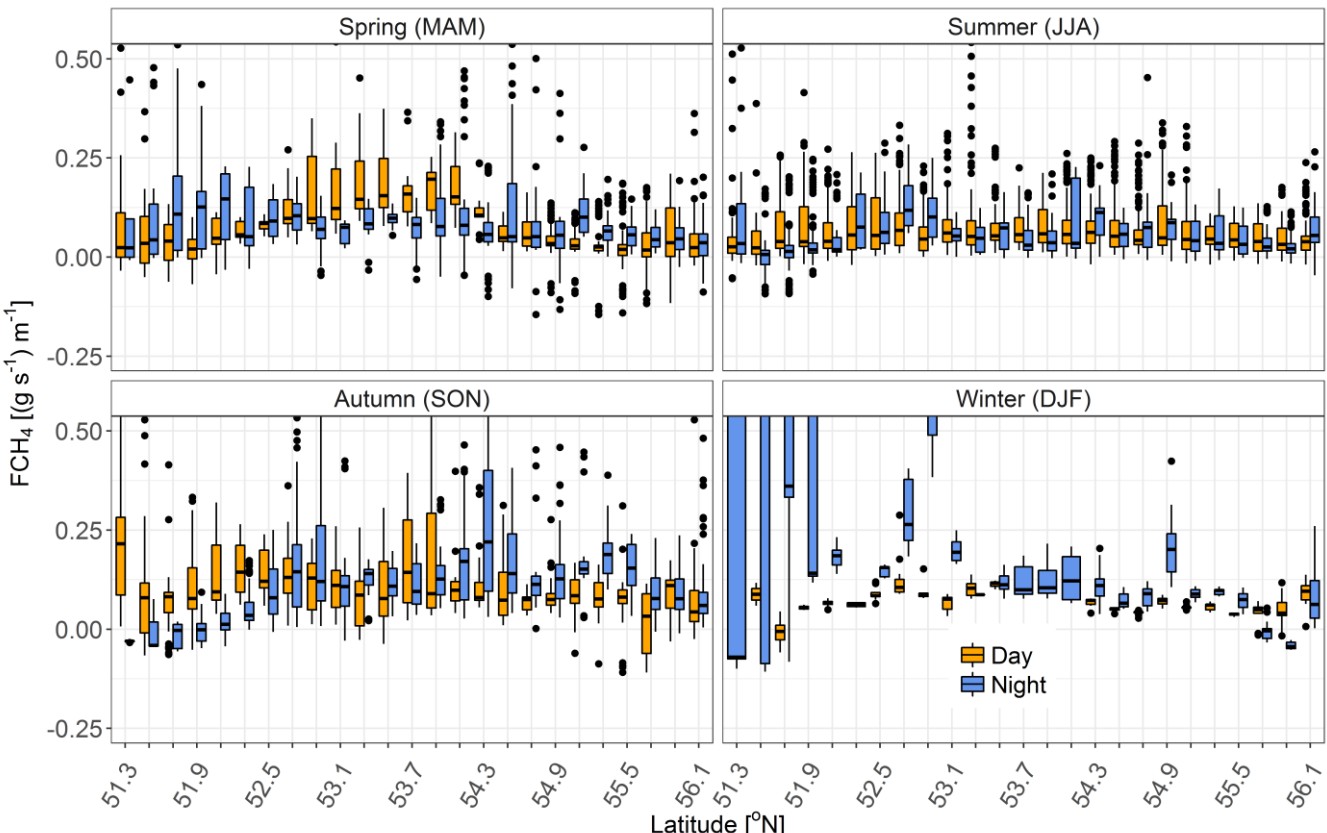

**Figure 9.** Box and whisker plots of 5-minute binned averages of $CH_4$ fluxes along the route of the ferry (latitude bin width: 0.2°) segregated into day and night contributions. The horizontal bar within each box corresponds to the median for a given latitude bin, the upper and lower hinges represent the 75th and 25th quantiles, respectively, and the upper and lower whiskers indicate the largest/smallest observation less/greater than or equal to upper/lower hinge +/- 1.5 * IQR (Inter-Quantile Range), respectively. The outliers are represented by solid circles and arithmetic means by red diamonds. The flux is integrated over the height of the planetary boundary layer and expressed in units of mass flux per meter travelled crosswind within each latitude bin per unit time.

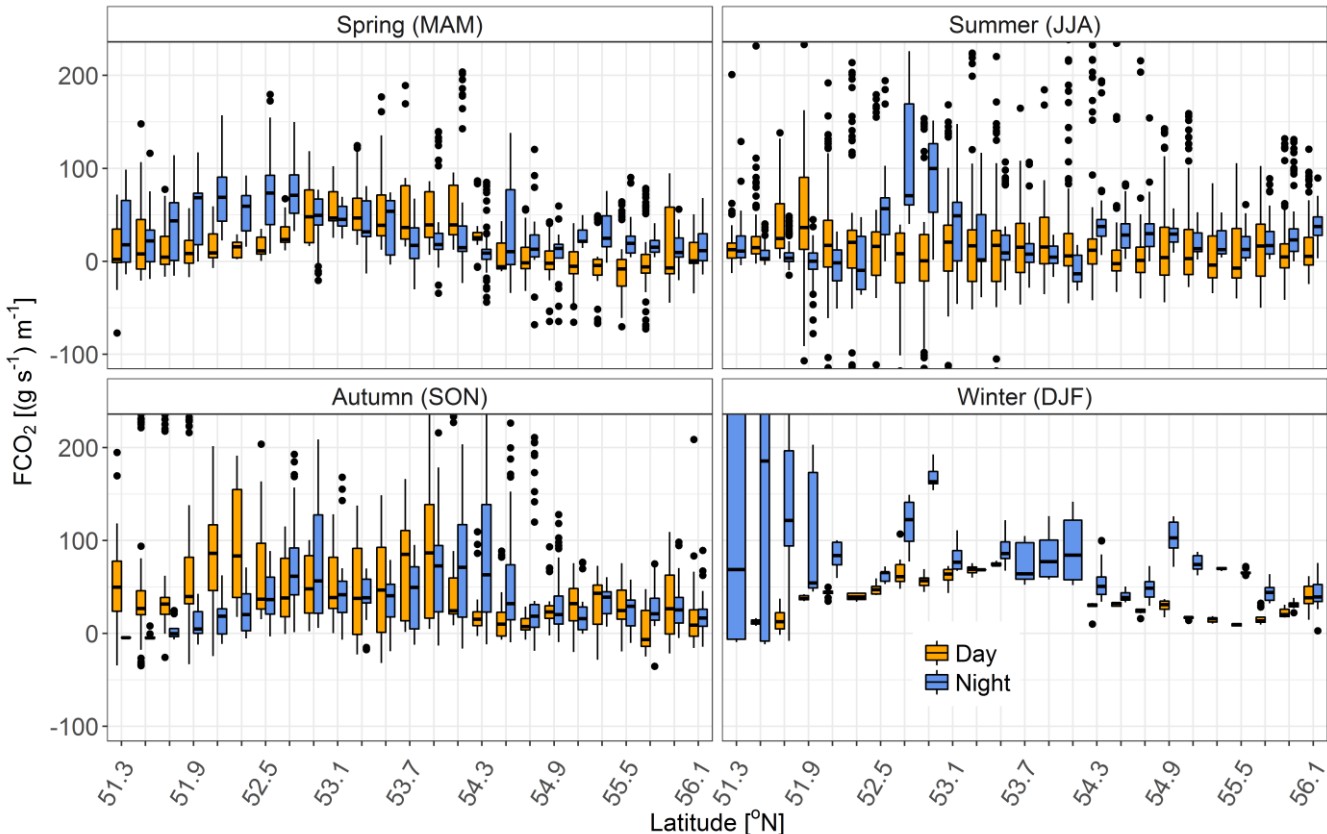

**Figure 10.** Box and whisker plots of 5-minute binned averages of $CO_2$ fluxes along the route of the ferry (latitude bin width: 0.2°) segregated into day and night contributions. The horizontal bar within each box corresponds to the median for a given latitude bin, the upper and lower hinges represent the 75th and 25th quantiles, respectively, and the upper and lower whiskers indicate the largest/smallest observation less/greater than or equal to upper/lower hinge +/- 1.5 * IQR (Inter-Quantile Range), respectively. The outliers are represented by solid circles and arithmetic means by red diamonds. The flux is integrated over the height of the planetary boundary layer and expressed in units of mass flux per meter travelled crosswind within each latitude bin per unit time.

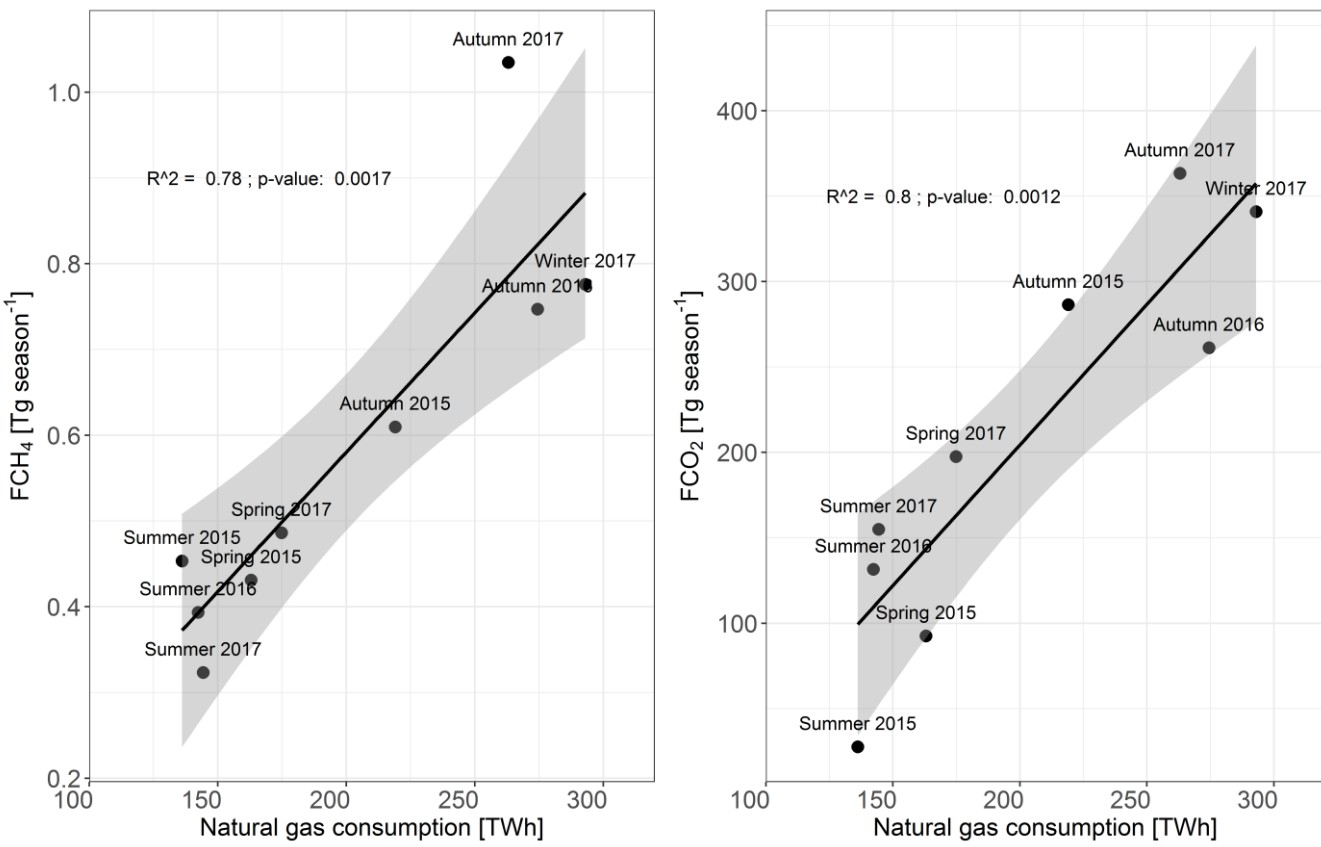

**Figure 11.** Seasonal budgets of CH₄ and CO₂ as function of UK natural gas consumption (source: BEIS, 2018). The shaded areas represent the 95% confidence intervals of the linear regressions.

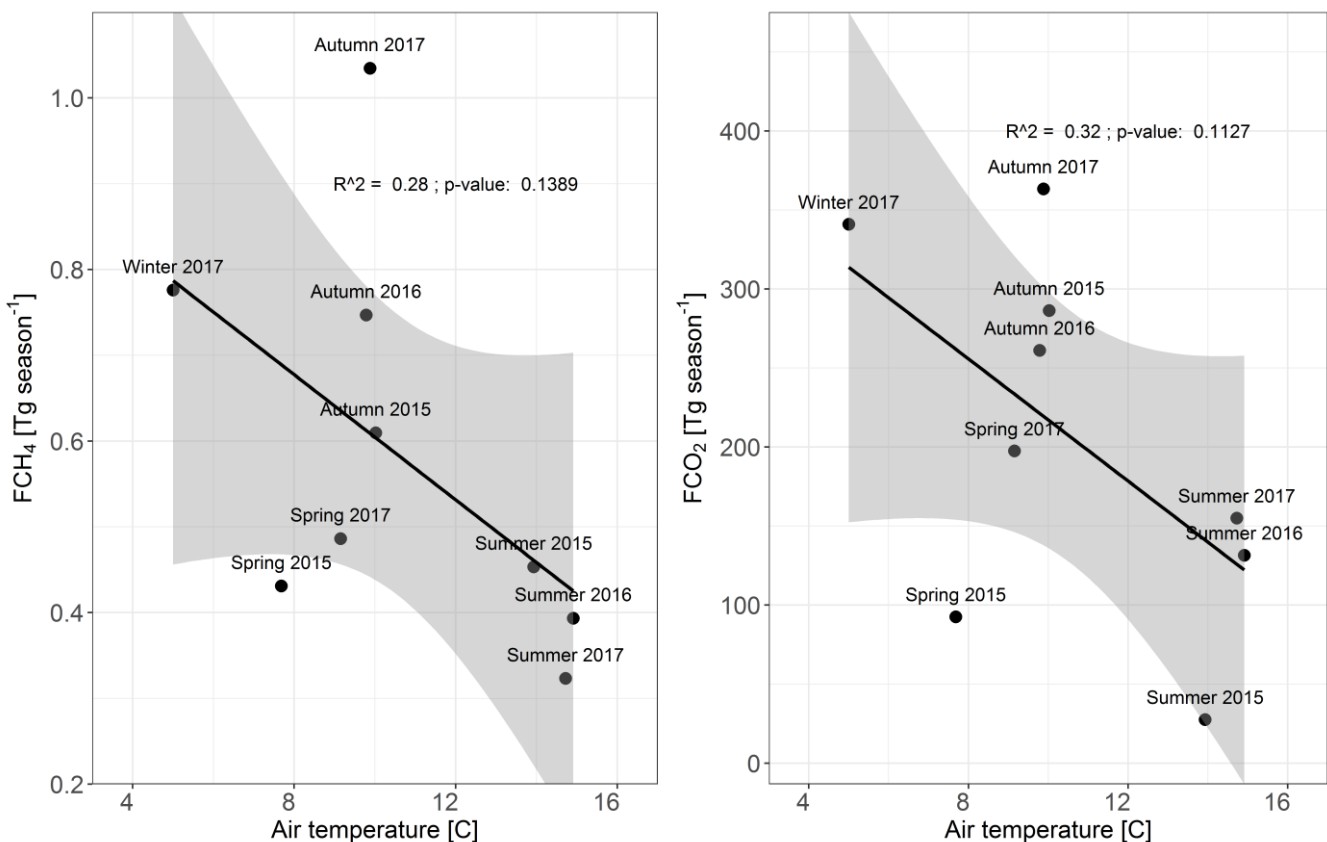

**Figure 12.** Seasonal budgets of $CH_4$ and $CO_2$ as function of mean UK air temperature derived from ca. 250 synoptic stations (source: Met Office, 2018). The shaded areas represent the 95% confidence interval of the linear regressions.

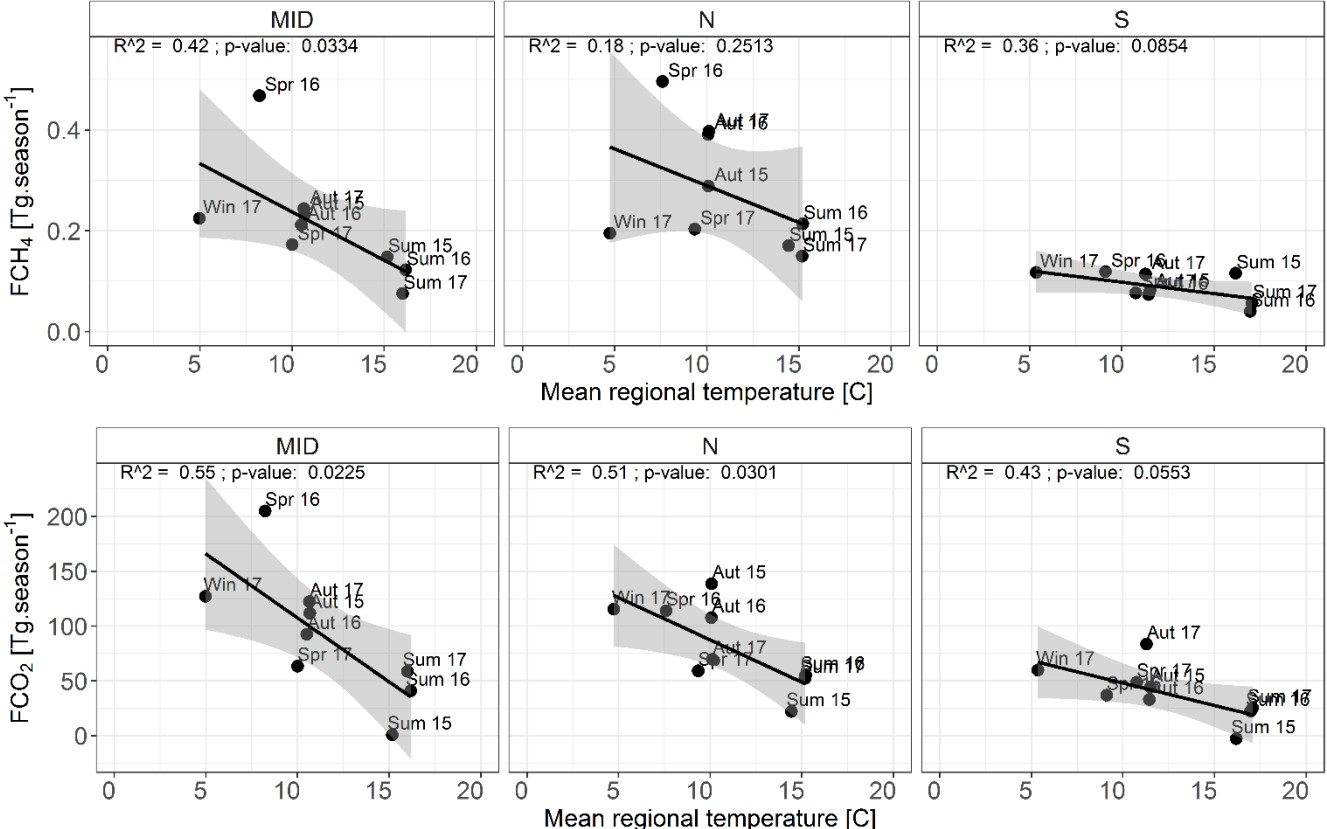

**Figure 13.** Seasonal fluxes of CH₄ and CO₂ estimated by mass balance from concentration measurements on board the ferry as function of mean regional air temperature (source: Met Office, 2018). The data are presented for three latitudinal regions denoted "MID", "N" and "S". The "MID" region spans the latitude range 52.8 °N – 54.2 °N, "N" spans 54.2 °N – 56.1 °N and "S" spans 52.0 °N – 52.8 °N. The shaded areas represent the 95% confidence intervals of the linear regressions.

**Table 1.** List of instruments and observables recorded on board the Rosyth (Scotland, UK; 56° 1' 21.611'' N, 3° 26' 21.558'' W) to Zeebrugge (Belgium; 51° 21' 16.96'' N, 3° 10' 34.645'' E) freight ferry.

| Observable | Unit | Instrument |
|---|---|---|
| $CO_2$ dry mole fraction | ppm | Picarro 1301 CRDS |
| $CH_4$ dry mole fraction | ppb | Picarro 1301 CRDS |
| Apparent wind speed (with respect to moving ship) | $m.s^{-1}$ | Vaisala WXT520 |
| Apparent wind direction (wind blowing from with respect to prow of moving ship) | degrees | Vaisala WXT520 |
| Air temperature | °C | Vaisala WXT520 |
| Ambient pressure | hPa | Vaisala WXT520 |
| Relative humidity | % | Vaisala WXT520 |
| Ship speed | kt | Garmin 18x series GPS |
| Ship bearing | degrees | Garmin 18x series GPS |
| Ship position, latitude | decimal | Garmin 18x series GPS |
| Ship position, longitude | decimal | Garmin 18x series GPS |

**Table 2.** Details of the weekly auto-calibration sequence (interval 169 hours) and reference gases. The references gases were calibrated by the Swiss Federal Laboratories for Materials Testing and Research (EMPA, Dübendorf, Switzerland) using a Picarro 1301 CRDS. Calibrations scales (NOAA/ ESRL): WMO-CH$_4$-X2004 for methane and WMO-CO$_2$-X2007 for carbon dioxide. The measurement uncertainties correspond to the standard deviation multiplied by a coverage factor k = 2, which provides a level of confidence of approximately 95 %.

| Step | Type | Time interval [s] | Calibration standard number | CO$_2$ ± uncertainty [ppm] | CH$_4$ ± uncertainty [ppb] |
|------|------|-------------------|-----------------------------|----------------------------|----------------------------|
| 1 | Purge | 300 | 1 | 384.23 ± 0.15 | 1815.36 ± 1.45 |
| 2 | Measurement | 900 | 1 | 384.23 ± 0.15 | 1815.36 ± 1.45 |
| 3 | Purge | 300 | 2 | 418.29 ± 0.16 | 2018.06 ± 1.58 |
| 4 | Measurement | 900 | 2 | 418.29 ± 0.16 | 2018.06 ± 1.58 |
| 5 | Purge | 300 | 3 | 474.86 ± 0.18 | 2426.77 ± 1.86 |
| 6 | Measurement | 900 | 3 | 474.86 ± 0.18 | 2426.77 ± 1.86 |
| 7 | Purge | 300 | Ambient air | Ambient air | Ambient air |

**Table 3.** Seasonal and annual budgets for $CO_2$ and $CH_4$ for the United Kingdom (excluding Scotland) and Ireland estimated by a mass balance approach using concentrations measured at the Mace Head station (Republic of Ireland; 53° 19' 19.2'' N, 9° 54' 3.599'' W) and on board the freight ferry which serves the Rosyth (Scotland, UK; 56° 1' 21.611'' N, 3° 26' 21.558'' W) to Zeebrugge (Belgium; 51° 21' 16.96'' N, 3° 10' 34.645'' E) route. Seasonal budgets were calculated by year – where sufficient data was available; seasonal budgets were also derived using the entire dataset with and without segregation of the raw fluxes into day and night components. Annual budgets were calculated with and without seasonality and with and without day/night segregation. The variability and uncertainty terms were calculated using Eq. 4 and Eq. 7-8, respectively.

| Season | Year | Flux ± uncertainty (variability) [Tg] | |
|---|---|---|---|
| | | $CO_2$ | $CH_4$ |
| Winter | 2015 | - | - |
| Spring | 2015 | 92.6 ± 21.1 (34.7) | 0.43 ± 0.13 (0.11) |
| Summer | 2015 | 27.6 ± 79.5 (46.8) | 0.45 ± 0.72 (0.09) |
| Autumn | 2015 | 286.4 ± 35.4 (47.6) | 0.61 ± 0.07 (0.14) |
| Winter | 2016 | - | - |
| Spring | 2016 | - | - |
| Summer | 2016 | 131.6 ± 82.6 (36.5) | 0.39 ± 0.25 (0.09) |
| Autumn | 2016 | 261.3 ± 164.3 (56.4) | 0.75 ± 0.40 (0.16) |
| Winter | 2017 | 341 ± 17.2 (62.1) | 0.78 ± 0.05 (0.38) |
| Spring | 2017 | 197.5 ± 40.4 (27.9) | 0.49 ± 0.14 (0.07) |
| Summer | 2017 | 155 ± 81.8 (77.6) | 0.32 ± 0.14 (0.06) |
| Autumn | 2017 | 363.4 ± 12.1 (65.7) | 1.03 ± 0.04 (0.15) |
| Winter | 2016 & 2017 | 379.1 ± 26.6 (68.8) | 0.89 ± 0.08 (0.35) |
| Spring | 2015 – 2017 | 161.5 ± 30.9 (41.2) | 0.55 ± 0.08 (0.17) |
| Summer | 2015 – 2017 | 123.6 ± 76.9 (64.6) | 0.38 ± 0.25 (0.09) |
| Autumn | 2015 – 2017 | 250.2 ± 200.1 (57.8) | 0.72 ± 0.40 (0.16) |
| Winter (day/night weighting) | 2016 & 2017 | 357.8 ± 26.2 (66.8) | 0.82 ± 0.08 (0.34) |
| Spring (day/night weighting) | 2015 – 2017 | 162.5 ± 30.9 (55.0) | 0.57 ± 0.08 (0.22) |
| Summer (day/night weighting) | 2015 – 2017 | 127.7 ± 76.9 (78.7) | 0.39 ± 0.25 (0.12) |
| Autumn (day/night weighting) | 2015 – 2017 | 232.9 ± 57.8 (72.2) | 0.67 ± 0.16 (0.19) |
| Annual (from seasonal budgets) | 2015 – 2017 | 914.4 ± 218.1 (118.1) | 2.55 ± 0.48 (0.43) |
| Annual (from seasonal, day/night weighted budgets) | 2015 – 2017 | 881.0 ± 125.8 (137.5) | 2.44 ± 0.30 (0.47) |
| Annual (no seasons) | 2015 – 2017 | 708.3 ± 270.4 (241.9) | 2.1 ± 0.67 (0.63) |
| Annual (no seasons, day/night weighted) | 2015 – 2017 | 598.3 ± 250.1 (274.9) | 1.66 ± 0.60 (0.94) |

| | | | |
|---|---|---|---|
| UK (Department for Business, 2017) | 2015 | 415.1 | 2.1 |
| RoI (Agency, 2017) | 2015 | 38.4 | 0.53 |
| Scotland (Inventory, 2018) | 2015 | 30.8 | 0.34 |
| Total inventory (UK – Scotland + RoI) | 2015 | 422.7 | 2.29 |
| Ganesan (Ganesan et al., 2015) | 2012 - 2014 | - | 1.65 - 2.67 |
| Bergamaschi (Bergamaschi et al., 2015) | 2006 - 2007 | | 3.1 – 3.5 |