# Peer review of "Country-scale greenhouse gases budgets using shipborne measurements: a case study for the United Kingdom and Ireland."

_Atmospheric Chemistry and Physics, 2018_

## Referee Comment (RC1) · Anonymous Referee #1 · 8 Nov 2018

Summary/General comments: Helfter et al. present observationally derived fluxes of $CO_2$ and $CH_4$ for the United Kingdom and Ireland using multiple years of shipborne observations and a mass balance approach. The authors have collected a unique dataset in making ferry based measurements downwind of an island nation and this has facilitated the application of a simple mass balance approach to evaluate national level fluxes of greenhouse gases. This is an interesting and valuable contribution to the scientific literature and well placed in ACP. I do have some questions and points of clarification that are needed, in particular some expanded discussion of uncertainty, and these are highlighted below. Once these issues have been addressed publication in ACP would be warranted.

[Figure]

Major comments: My largest complaint is that I would like to see more explanation and detailed information on the uncertainty assessment. While uncertainties are derived for all the mass balance estimates, I would like to see more explanation of how these were derived and inclusion of quantitative measures of uncertainty for all the components that make up the final uncertainty. I would also like to see discussion of what is included and what possible errors are excluded from these uncertainties. One issue in particular I would like to see more on is the estimates and assumptions of planetary boundary layer depth. In the mass balance approach applied there are assumptions of tracers being well-mixed in the planetary boundary layer and the height of this layer remains stable in time. First, it is better to use the term mixed layer as though often synonymous with pbl it isn't always the same. My understanding is the mixing layer was extracted from WRF and averaged over the history of the particle travel. What were these values? How much variance in pbl depth occurred in this time (as a proxy for uncertainty). How were these pbl heights evaluated and how do we know a bias isn't present? How does the model do at representing mixing layers over the ocean, where the measurements were made? Are local sea-breeze effects important? Is the pbl not changing over the course of the full downwind transect of the boat? If it is changing how is that dealt with? How can observations be safely used at any time of day? In airborne mass balance observation need to be midday when sufficient mixing within the pbl can be established and the pbl thickness is not changing dramatically?

So I would like to see generally more information on uncertainty, and specifically a more detailed discussion, quantitative analysis, explanation and justification of the approach regarding mixing depths and assumptions therein.

Minor comments: Page 2 Line 9-11: This is misleading as a number of other papers have shown the possible role of OH (Turner et al., PNAS; Rigby et al., PNAS) and roles of fossil/fires (for example Wordern et al., GRL).

Section 2.1.1 Is there filtering of potential contamination of stack air and/or passenger influenced air? If not, how justified? If so, how is the filtering done?

[Figure]

Was the H2O correction for the Picarro validated/calibrated?

Section 2.2.3 See above regarding PBL questions

Page 5 Line 35: Please tell us what these mean travel times are and their variance.

3.2 Diurnal variability: Are these defined by observation time or by flux time? I'm concerned about the use of data not during well mixed conditions, and what transects may show during time of pbl growth and collapse. Also, how do we know that night measurements are not missing a large outflow occurring above the stable surface layer?

Inventories are mentioned but never really discussed in any detail. More explanation of the inventories and what is included/excluded would be valuable.

Page 8 line 18: repeating point c

Page 10 line 12 : state here explicitly this implies the biosphere accounts for the difference as opposed to saying cannot compare with the inventory. Could also mention with biospheric model and/or constraints could directly compare.

―――――――――――――――――

---

## Referee Comment (RC2) · Anonymous Referee #2 · 10 Dec 2018

This paper reports the seasonal and annual budgets of CO2 and CH4 of the United Kingdom and Ireland over a period of three years from 2015 – 2017 using a mass balance approach and mixing ratio measurements on board a ferry that traversed the eastern side of the measurement site. I think that what is noteworthy about this contribution is the unique data set obtained from long-term measurements that enabled the seasonal and annual estimation of the fluxes, which is not possible from snapshot measurements using an aircraft –based platform. I also commend the authors for the rigor with which they estimated the fluxes using various approaches (considering seasonality, daytime/nighttime differences, etc.). The paper is also generally well-written with only a few typographical and grammatical mistakes.

[Figure]

I did notice, however, that the background concentration of CO2 and CH4 had so much variability, and that the authors used a mathematical fitting routine to obtain a smoothened background signal. Based on previous studies using the mass balance approach, the large variability in the background actually contributes a significant variability (i.e. uncertainty) in the estimated fluxes. Some of my thoughts on this are outlined below in the specific comments.

Specific comments:

(1) Not all your readers will be familiar with the geography of the measurement domain. Please provide a map of your measurement site together with the path of the ferry used in the study. Please label the map with the cities for reference. Direction of prevailing winds throughout the season will be also useful so that the reader can clearly see the transect of the ferry relative to the prevailing winds. This should be your figure 1. It helps if you set the stage for your readers.

(2) Figure 1 already shows the Hysplit backward trajectory frequencies but the reader will just assume that the ship is located where the highest frequency is found (color red). Also, the two rows apparently show two succeeding days in May 2015. That is not clearly described in the figure caption and the reader discovers this only after staring at the figure. I think that this figure should not be your first figure.

(3) Page 3: You state that background measurements were taken at the Mace Head site in Ireland – all the more reason why your figure 1 should include what's stated above in (1) but also the location of the background site. It's important to set the stage for the reader for greater appreciation of the measurements and the analysis.

(4) Page 3: Your figure 1 should be something like your Figure 2 but with more detail such as an arrow that shows the wind direction. It is likely not westerly winds throughout the year. It will be instructive if the authors are deliberate about stating/describing the meteorological conditions throughout the year. It would be good to show a windrose plot to support this.

(5) page 3 line 16: Rosynth – Zeebrugge are two important ferry end points that should be included in Figure 1.

(6) Page 3, line 30: Please state in the text where the calibration gases were obtained, how many was used, and their mixing ratios. Did they span the expected range of measured CO2 and CH4 mixing ratios from the measurement site? I note that the calibration gas was measured for 15 minutes? How often do you obtain a data point? Every 1 minute, every 15 seconds? Please state. How did you obtain the mean and uncertainty that was reported in Table 2, i.e. how many data points did you average every 15 minutes? What is the coverage factor k that was mentioned in Table 2. Please explain. And did you purge with the reference gas or with zero air?

(7) Page 4 and also page 5: Assumptions: What is tricky about the mass balance approach is the choice/estimation of the background. It can contribute one of the largest uncertainty in the estimation of the emission flux – because the air mass travels a couple of hours from the background site to the receptor site. Your measurements at the Mace Head site showed significant variability in the background. What's the effect of this variability on the estimated flux?

It is possible that the uncertainty is significantly larger than what equation (4) is estimating simply because there is so much variability in the background mixing ratio of CO2 and CH4. Based on the results of previous studies, the variability in the background significantly contributes to the uncertainty in the estimated fluxes. The authors are then advised to do a sensitivity analysis of the obtained fluxes using the standard deviations in the background obtained from Mace Head and comment on the results in the discussion (comparing against the uncertainty obtained in equation 4). Furthermore, on page 5 regarding the mass balance approach, what was the time - shift interval that was used to obtain the background? I ask this question because there is a significant travel time for the air mass to reach the receptor site. This is also the reason why in previous aircraft-based mass balance approaches, the mixing ratios at the "wings" of the transect (outside the plume) were actually used as the baseline or background

mixing ratio. (8) Page 4 on Data screening. Please state in this section that the histogram of data points showed in Figure 3 correspond to the data points that satisfy the data screening protocol, which emphasizes that only the data with westerly winds were used in the averaging and flux calculations.

Out of all the data points that you collected, how many points were used in the analysis? What's the percentage of useful data? Are there more points for certain months relative to others?

(9) Page 5: The authors used the wind speed and wind direction from the WRF model to calculate the fluxes. I am sure that there are multiple synoptic stations in UK and Ireland? How come the wind speed and wind direction data from those synoptic stations were not used in this study?

Were the back trajectories consistent with the synoptic station data? At what height above the ground were the back trajectories modeled for?

It will be good to show the time series of the PBL heights in the supplementary information for the domain and for the period of measurement. It would be also good to report if the modeled PBL depth has been validated with previous measurements (at least for those previous years when measurements of PBL depths were available), just to reassure yourselves that the modeled values are sufficient to be used in your calculations.

(10) Results. On seasonal and annual fluxes. Please check the units of your fluxes in Figure 6 and 7. I believe you meant g s-1 m-2 rather than g s-1 m-1.

Please also report the total number of data points per season.

(11) Figure 10 and 11. What does the gray shade represent in the figures? Please explain in the figure caption.

In Figure 11, you used the mean air temperature over the measurement domain. How many synoptic stations were used when you averaged the air temperature?

---

## Author Comment (AC1) · 23 Jan 2019

We thank the referee for the thorough and fair assessment of our manuscript. We have addressed all comments and a complete response is provided in the accompanying file. Also included is a new Supplementary Material document.

Please also note the supplement to this comment: https://www.atmos-chem-phys-discuss.net/acp-2018-948/acp-2018-948-AC1-supplement.zip

2018.

---

## Author Response (AR1)

**Authors' response to comments by anonymous referee #1 on Atmos. Chem. Phys. Discuss., manuscript acp-2018-948, "Country-scale greenhouse gases budgets using shipborne measurements: a case study for the United Kingdom and Ireland." by Helfter et al.**

We thank the referee for recognising the uniqueness of the dataset presented in this paper and for recommending its publication in ACP.

The referee's comments are set in bold, our responses in italics and new text, quoted from the revised manuscript is highlighted in blue.

**Major comments**

- **More explanation of how uncertainties were derived and inclusion of quantitative measures of uncertainty for all the components that make up the final uncertainty.**

*The following methods paragraph (blue text) on uncertainties an error propagation was added to section 2.2.4 in order to provide an explicit treatment of the uncertainties on the calculated budgets.*

Uncertainty and error propagation

In addition to the temporal variability $\Delta F_c$, (Eq. 4) we calculated the uncertainty on the total fluxes arising from the uncertainties on the individual terms of the mass balance equation. Noting that $dx$ represents the distance travelled by the ship with speed $v_{ship}$ during the infinitesimal time interval $dt$, Eq. 2 can be reformulated to express the partial flux $f_c$ through a 2-dimensional plane spanning the horizontal distance $dx$ as a function of $v_{ship}$ and $dt$ (Eq. 5).

$$f_C = \int_{z_{ground}}^{z_{PBL}} \Delta\chi_C \cdot U \cdot n_{air}(z) \cdot \cos\theta \cdot v_{ship}\ dt\ dz \tag{5}$$

Applying the rules of error propagation, the error on the flux term $f_c$ ($\delta f_c$) is given by (with $N_{air}$, the value of the integral of $n_{air}(z)$ evaluated over time step $dt$):

$$\frac{\delta f_c}{|f_c|} = \sqrt{\left(\frac{\delta\chi_c}{\chi_c}\right)^2 + \left(\frac{\delta U}{U}\right)^2 + \left(\frac{\delta\cos\theta}{\cos\theta}\right)^2 + \left(\frac{\delta v_{ship}}{v_{ship}}\right)^2 + \left(\frac{\delta dt}{dt}\right)^2 + \left(\frac{\delta \int_{z_{ground}}^{z_{PBL}} n_{air}(z)\ dz}{N_{air}}\right)^2} \tag{6}$$

Assuming that, (a) the uncertainty on $dt$ is negligible, and (b) the uncertainty on the PBL height ($z_{PBL}$) is the dominant error term in the integral of $n_{air}(z)$ between height $z_{ground}$ and $z_{PBL}$, Eq. 6 can be approximated as:

$$\frac{\delta f_c}{|f_c|} \approx \sqrt{\left(\frac{\delta\chi_c}{\chi_c}\right)^2 + \left(\frac{\delta U}{U}\right)^2 + \left(\frac{\delta\cos\theta}{\cos\theta}\right)^2 + \left(\frac{\delta v_{ship}}{v_{ship}}\right)^2 + \left[\frac{\left(n_{air}(z_{PBL}) - n_{air}(z_{ground})\right) \cdot \delta z_{PBL}}{N_{air}}\right]^2} \tag{7}$$

Finally, similarly to Eq. 4, the total error on the flux $F_c$ ($\delta F_c$) calculated for a complete transect of the ship between $x_{min}$ and $x_{max}$ is given by:

$$\frac{\delta F_c}{|F_c|} = \sqrt{\sum_i^N \left\{\left(\frac{\delta f_c}{|f_c|}\right)_i\right\}^2} \tag{8}$$

The standard deviations of the individual terms in Eq. 7, calculated for each 5-minute averaging period and averaged over each nominal latitude bin, were used as proxies for uncertainties.

*For disambiguation, the terminology used in the previous version of the manuscript was changed from "uncertainty" to "variability" throughout the document. In the revised manuscript, the term uncertainty denotes the error propagated using Eq. 7 and 8. Table 3 was updated and now provides both budget variability and uncertainty.*

**Table 3.** Seasonal and annual budgets for $CO_2$ and $CH_4$ for the United Kingdom (excluding Scotland) and Ireland estimated by a mass balance approach using concentrations measured at the Mace Head station (Republic of Ireland; 53° 19' 19.2'' N, 9° 54' 3.599'' W) and on board the freight ferry which serves the Rosyth (Scotland, UK; 56° 1' 21.611'' N, 3° 26' 21.558'' W) to Zeebrugge (Belgium; 51° 21' 16.96'' N, 3° 10' 34.645'' E) route. Seasonal budgets were calculated by year – where sufficient data was available; seasonal budgets were also derived using the entire dataset with and without segregation of the raw fluxes into day and night components. Annual budgets were calculated with and without seasonality and with and without day/night segregation. The variability and uncertainty terms were calculated using Eq. 4 and Eq. 7-8, respectively.

| Season | Year | Flux ± uncertainty (variability) [Tg] | |
|---|---|---|---|
| | | $CO_2$ | $CH_4$ |
| Winter | 2015 | - | - |
| Spring | 2015 | 92.6 ± 21.1 (34.7) | 0.43 ± 0.13 (0.11) |
| Summer | 2015 | 27.6 ± 79.5 (46.8) | 0.45 ± 0.72 (0.09) |
| Autumn | 2015 | 286.4 ± 35.4 (47.6) | 0.61 ± 0.07 (0.14) |
| Winter | 2016 | - | - |
| Spring | 2016 | - | - |
| Summer | 2016 | 131.6 ± 82.6 (36.5) | 0.39 ± 0.25 (0.09) |
| Autumn | 2016 | 261.3 ± 164.3 (56.4) | 0.75 ± 0.40 (0.16) |
| Winter | 2017 | 341 ± 17.2 (62.1) | 0.78 ± 0.05 (0.38) |
| Spring | 2017 | 197.5 ± 40.4 (27.9) | 0.49 ± 0.14 (0.07) |
| Summer | 2017 | 155 ± 81.8 (77.6) | 0.32 ± 0.14 (0.06) |
| Autumn | 2017 | 363.4 ± 12.1 (65.7) | 1.03 ± 0.04 (0.15) |
| Winter | 2016 & 2017 | 379.1 ± 26.6 (68.8) | 0.89 ± 0.08 (0.35) |
| Spring | 2015 – 2017 | 161.5 ± 30.9 (41.2) | 0.55 ± 0.08 (0.17) |
| Summer | 2015 – 2017 | 123.6 ± 76.9 (64.6) | 0.38 ± 0.25 (0.09) |
| Autumn | 2015 – 2017 | 250.2 ± 200.1 (57.8) | 0.72 ± 0.40 (0.16) |
| Winter (day/night weighting) | 2016 & 2017 | 357.8 ± 26.2 (66.8) | 0.82 ± 0.08 (0.34) |
| Spring (day/night weighting) | 2015 – 2017 | 162.5 ± 30.9 (55.0) | 0.57 ± 0.08 (0.22) |
| Summer (day/night weighting) | 2015 – 2017 | 127.7 ± 76.9 (78.7) | 0.39 ± 0.25 (0.12) |
| Autumn (day/night weighting) | 2015 – 2017 | 232.9 ± 57.8 (72.2) | 0.67 ± 0.16 (0.19) |
| Annual (from seasonal budgets) | 2015 – 2017 | 914.4 ± 218.1 (118.1) | 2.55 ± 0.48 (0.43) |
| Annual (from seasonal, day/night weighted budgets) | 2015 – 2017 | 881.0 ± 125.8 (137.5) | 2.44 ± 0.30 (0.47) |
| Annual (no seasons) | 2015 – 2017 | 708.3 ± 270.4 (241.9) | 2.1 ± 0.67 (0.63) |

| | | | |
|---|---|---|---|
| Annual (no seasons, day/night weighted) | 2015 – 2017 | 598.3 ± 250.1 (274.9) | 1.66 ± 0.60 (0.94) |
| UK (Department for Business, 2017) | 2015 | 415.1 | 2.1 |
| RoI (Agency, 2017) | 2015 | 38.4 | 0.53 |
| Scotland (Inventory, 2018) | 2015 | 30.8 | 0.34 |
| Total inventory (UK – Scotland + RoI) | 2015 | 422.7 | 2.29 |
| Ganesan (Ganesan et al., 2015) | 2012 - 2014 | - | 1.65 - 2.67 |
| Bergamaschi (Bergamaschi et al., 2015) | 2006 - 2007 | | 3.1 – 3.5 |

*Finally, the relative contribution of each uncertainty term was evaluated for each season and year of the study and provided in Table S2 of the Supplementary Material document.*

Table S2: Relative contribution of the individual uncertainty terms to the total uncertainty and total uncertainty on the calculated emissions budgets per season and year of the study. The difference between time-lagged and instantaneous emissions budgets illustrates the impact of factoring in the mean West-to-East air mass travel time in the selection of the reference concentrations measured at Mace Head.

| Season | Year | Relative contribution to total uncertainty [%] | | | | | Total uncertainty on emissions budget [%] | | Difference between time-lagged and instantaneous emissions budgets [%] | | Mean air mass travel time ± SD [hour] |
|---|---|---|---|---|---|---|---|---|---|---|---|
| | | Wind speed in PBL | Molar density | Mole fraction (enhancement above background) | Projection angle θ | Ship speed | $CO_2$ | $CH_4$ | $CO_2$ | $CH_4$ | |
| Winter | 2015 | - | - | - | - | - | - | - | - | - | 14.7 ± 4.7 |
| Spring | 2015 | 26 | 4 | 67 | 0 | 2 | 23 | 30 | 2.6 | 0.9 | 15.8 ± 5.0 |
| Summer | 2015 | 39 | 3 | 54 | 1 | 3 | 288 | 160 | 14.5 | 0.3 | 23.1 ± 9.9 |
| Autumn | 2015 | 48 | 5 | 43 | 2 | 2 | 74 | 11 | 2.0 | 0.3 | 15.8 ± 5.0 |
| Winter | 2016 | - | - | - | - | - | - | - | - | - | 15.2 ± 0.5 |
| Spring | 2016 | - | - | - | - | - | - | - | - | - | 14.7 ± 2.6 |
| Summer | 2016 | 45 | 4 | 49 | 1 | 2 | 63 | 64 | 1.4 | 0.5 | 20.2 ± 8.8 |
| Autumn | 2016 | 31 | 3 | 63 | 1 | 2 | 63 | 53 | 0.2 | 0.2 | 16.4 ± 7.4 |
| Winter | 2017 | 80 | 7 | 8 | 1 | 4 | 5 | 6 | 0.4 | 0.2 | 13.5 ± 4.1 |
| Spring | 2017 | 62 | 7 | 26 | 1 | 4 | 20 | 29 | 0.5 | 0.1 | 16.4 ± 6.2 |
| Summer | 2017 | 44 | 4 | 49 | 1 | 2 | 53 | 44 | 2.2 | 0.2 | 18.3 ± 6.4 |
| Autumn | 2017 | 71 | 4 | 20 | 1 | 4 | 3 | 4 | 0.9 | 0.2 | 15.9 ± 4.8 |

▪ **Discussion of what is included and what possible errors are excluded.**

*A paragraph was added at the beginning of section 4 (Discussion) which discusses the assumptions underpinning the mass balance approach and potential bias arising from using PBL height and wind speed extracted from WRF.*

[revised manuscript text omitted]

- **In particular, more detail on estimates and assumptions of mixed layer height:**
  - **Assumption of tracers being well-mixed: indeed but cannot be verified.**

*This comment is pertinent indeed, and we have added a comment to this effect in the new paragraph in section 4 (see above).*

- **Height of this layer remains constant in time:**

*The mean PBL height and associated standard deviation was calculated for hourly time intervals as outlined in the methods section 2.2.3 (also note new text in blue):*

"The WRF model hourly output from the UK domain was used to calculate spatial means and standard deviations of the wind speed, wind direction, and the planetary boundary layer height. We estimate the spatial averages at a height of ~450 m (4th model layer) for an area defined as follow: lower left corner coordinates of 52.0 latitude and -10.0 longitude and the upper right corner of 57.0 latitude and 3.0 longitude. Time series of hourly averages of wind speed, wind direction and PBL height were constructed for the data period 01/01/2014 to 31/12/2017."

- **Better to use mixed layer [than PBL]**

*We agree with the referee that PBL and mixed layer heights are often used interchangeably but since the values used in this study were derived from the WRF model, we opted to use the preferred terminology of the WRF modelling community which refers to PBL rather than mixed layer.*

- **My understanding is that the mixing layer was extracted from WRF and averaged over the history of the particle travel:**

*The mixing layer height was indeed extracted form WRF but on hourly intervals and independently of the particle travel time. This is outlined in section 2.2.3 quoted above.*

- **What were these values [of mixed layer height]:**

*The PBL height data extracted from WRF have now been summarised in Fig. S1 of the new Supplementary Material. Although hourly values were extracted from WRF, Fig. S1 presents daily means ± standard deviation for the sake of clarity.*

[Figure]

Figure S1: Daily mean PBL height (solid dots) and standard deviation (shaded ribbon) obtained by averaging the hourly values extracted from the WRF model with YSU scheme for the study period 2015-2017.

- **How much variance in PBL depth occurred in this time?**

*We hope that the daily standard deviations shown in Fig. S1 of the new Supplementary Material adequately answer the referee's question.*

- **How were these PBL heights evaluated: extracted from WRF. How do we know a bias [on PBL height] isn't present? How does the model do at representing mixing layers over the ocean, where the measurements were made?**
- *We have expanded Section 4 to discuss the uncertainty and bias between model output and observations, and we hope that the referee will find these comments sufficient. We also reviewed recent literature to answer the question about the representation of mixing layers over oceans. We refer the reader to the text quoted under bullet point **Discussion of what is included and what possible errors are excluded** in this document.*

- **Are local sea-breeze effects important?**

*Sea breeze effects are likely to be important. We have reviewed relevant recent literature and inserted comments regarding the treatment of sea breezes within WRF. We refer the reader to the text quoted under bullet point **Discussion of what is included and what possible errors are excluded** in this document.*

- **Is the PBL not changing over the course of the full downwind transect of the boat? If it is changing, how is that dealt with?**

*The PBL height did exhibit changes of the course of a full transect and this is why we opted to use hourly-averaged values of PBL heights.*

- **How can observations be safely used at any time of day?**

*Sufficient mixing throughout the PBL was hypothesised, although we acknowledge that the validity of this assumption might have been stretched at times (e.g. night time). The validity of and the uncertainty arising from this hypothesis of sufficient mixing could however not be tested nor quantified and we therefore accept it as a known unknown (this point is now explicitly discussed in Section 4; see quotation of new text under a previous comment regarding sources of errors).*

**Minor comments**

- **Page 2 line 9-11: This is misleading as a number of other papers have shown the possible role of OH (Turner et al., PNAS; Rigby et al., PNAS) and roles of fossil/fires (for example Wordern et al., GRL).**

*We agree with the referee's comment and the paragraph has been extended as follows (new text in blue):*
"At the global scale, total methane emissions from fossil fuels (from the fossil fuel industry and from geological seepage) have been relatively steady over the past three decades but research indicates that the estimates must be revised upwards by as much as 60%-110% (Schwietzke et al., 2016). Several mechanisms have been proposed to explain the recent rise in atmospheric methane; these include increases in emissions from microbial sources, which are meteorologically driven and can therefore exhibit substantial inter-annual variability (Dlugokencky et al., 2011; Nisbet, 2016; Schwietzke et al., 2016), a weakening of the hydroxyl (OH) chemical sink strength (Rigby et al., 2017; Turner et al., 2017) and an increase in fossil fuel contributions in the context of a stable OH sink and a downward revision of the biomass burning budget term (Worden et al., 2017)."

- **Section 2.1.1: Is there filtering of potential contamination of stack air and/or passenger influenced air? If not, how justified? If so, how is the filtering done?**

*Potentially contaminated data points due to on-board activities were indeed filtered out as part of the pre-processing data screening procedure described in section 2.2.1 but this filtering step was not explicitly described. We therefore added a bullet point before bullet point # 4 in section 2.2.1 which reads (new text in blue):*
"The relative wind direction measured on the ship (fixed reference point the prow of the vessel) was outside the range 150°-210°. This criterion was used to exclude data points potentially contaminated by on-board activities (e.g. emissions from chimney stacks)."

- **Was the H$_2$O correction for the Picarro validated/calibrated?**

*We relied on the manufacturer's own calibration and did not carry out an independent validation.*

- **Page 5 line 35: Please tell us what these mean travel times are and their variance.**

*The sentence has been extended to clarify this point:*
"The baseline mole fraction used to calculate the upwind enhancement of compound c were time-shifted in order to account for the mean air mass travel time across the domain (time taken to travel West-East from the longitude of the Mace Head station to the location of the ferry at hourly mean wind speed derived from the WRF model; see Table S2 of the Supplementary material for seasonal mean values and standard deviations)."
*See also Table S2 under an earlier comment in this document.*

- **3.2 Diurnal variability:**
  - **Are these defined by observation time or by flux time?**

*The discussion is done on the basis of observation time with an explicitly-stated caveat that "It is important to note the air mass transit time between the in- and out-flow points of the domain varied from a median of 11 hours in winter to 19 hours in summer, which means that the day and night periods did overlap". We consider this caveat to be sufficient and did therefore not add any additional comment.*

  - **Use of data during not well-mixed conditions and what transects may show during time of PBL growth and collapse. Also, how do we know that night time measurements are not missing a large outflow occurring above the stable surface layer?**

*Good mixing of the air column and absence of mass leakage and ingress into the 3D spatial domain are two of the simplifying assumptions outlined in section 2.2.; whilst they could not be tested on a point per point basis, we assume that a sufficient number of observations were made to bring the mixing-related uncertainties in line with*

*the overall measurement uncertainty. The discussion of uncertainties, including those arising from atmospheric mixing, has been extended, as detailed previously, and we hope that it answers this comment satisfactorily.*

- **Inventories are mentioned but never really discussed in any detail. More explanation of the inventories and what is included/excluded would be valuable.**

*A new reference has been added, which provides a full description of the methodology used for UK emissions mapping.*

BEIS (Department for Business, Energy and Industrial Strategy): UK Emission Mapping Methodology, available at https://uk-air.defra.gov.uk/assets/documents/reports/cat07/1812061112_MappingMethodology-for-NAEI-2016.pdf (last access 23 January 2019), 2016.

- **Page 8 line 18: repeating point c.**

*This has been changed to d).*

- **Page 10 line 12: state here explicitly this implies the biosphere accounts for the difference as opposed to saying cannot compare with the inventory. Could also mention with biospheric model and/or constraints could directly compare.**

*The point that biospheric emissions account for the difference between the mass balance budget for $CO_2$ and the inventory value is made on page 10 line 18. For this reason and also because we deemed it important to reiterate that the atmospheric inventory only considers anthropogenic sources, we opted not to change the wording of line 12.*

**New references**

Balzarini, A., Angelini, F., Ferrero, L., Moscatelli, M., Perrone, M. G., Pirovano, G., Riva, G.M., Sangiorgi, G., Toppetti, A.M., Gobbi, G.P., and Bolzacchini, E., Sensitivity analysis of PBL schemes by comparing WRF model and experimental data, Geosci. Model Dev. Discuss., 7, 6133–6171, www.geosci-model-dev-discuss.net/7/6133/2014/, doi:10.5194/gmdd-7-6133-2014, 2014.

Banks R.F., Tiana-Alsina J., Baldasano J.M., Rocadenbosch F., Papayannis A., Solomos S., Tzanis C.G., Sensitivity of boundary-layer variables to PBL schemes in the WRF model based on surface meteorological observations, LIDAR, and radiosondes during the HygrA-CD campaign, Atmos. Res., Vol. 176–177, pp. 185-201, https://doi.org/10.1016/j.atmosres.2016.02.024, 2016.

BEIS (Department for Business, Energy and Industrial Strategy): UK Emission Mapping Methodology, available at https://uk-air.defra.gov.uk/assets/documents/reports/cat07/1812061112_MappingMethodology-for-NAEI-2016.pdf (last access 23 January 2019), 2016.

Carslaw, D.C. and K. Ropkins, (2012) openair — an R package for air quality data analysis. Environ. Modell. Softw., Vol. 27-28, 52-61.

Hu, X.M., Nielsen-Gammon, J.W., Zhang, F., Evaluation of Three Planetary Boundary Layer Schemes in the WRF Model, J. Appl. Meteorol., 49(9), doi: 10.1175/2010JAMC2432.1, 2010.

Rigby, M., Montzka, S.A., Prinn, R.G., White, J.W.C., Young, D., O'Doherty, S., Lunt, M.F., Ganesan, A.L., Manning, A.J., Simmonds, P.G., Salameh, P.K., Harth, C.M., Mühle, J., Weiss, R.F., Fraser, P.J., Steele, L.P., Krummel, P.B., McCulloch, A., and Park, S., Role of atmospheric oxidation in recent methane growth, P. Natl. Acad. Sci. USA, 114 (21) 5373-5377; doi:10.1073/pnas.1616426114, 2017.

Salvador, N., Ayres, A. G., Santiago, A., Albuquerque, T.T.A., Neyval C. Reis Jr., Santos, J. M., Landulfo, E., Moreira, G., Lopes, F., Held, G., and Moreira, D.M.: Study of the Thermal Internal Boundary Layer in Sea Breeze Conditions Using Different Parameterizations: Application of the WRF Model in the Greater Vitória Region, Rev. Bras. Meteorol., *31*(4, Suppl. 1), 593-609. https://dx.doi.org/10.1590/0102-7786312314b20150093, 2016.

Steele, C. J., Dorling, S. R., von Glasow, R., and Bacon, J.: Idealized WRF model sensitivity simulations of sea breeze types and their effects on offshore windfields, Atmos. Chem. Phys., 13, 443-461, https://doi.org/10.5194/acp-13-443-2013, 2013.

Steele, C. J., Dorling, S. R., von Glasow, R. and Bacon, J., Modelling sea-breeze climatologies and interactions on coasts in the southern North Sea: implications for offshore wind energy. Q.J.R. Meteorol. Soc., 141: 1821-1835. doi:10.1002/qj.2484, 2015.

Turner, A.J., Frankenberg, C., Wennberg, P.O., and Jacob, D.J., Ambiguity in the causes for decadal trends in atmospheric methane and hydroxyl, P. Natl. Acad. Sci. USA, 114 (21) 5367-5372; doi:10.1073/pnas.1616020114, 2017.

Tyagi, B., Magliulo, V., Finardi, S., Gasbarra, D., Carlucci, P., Toscano, P., Zaldei, A., Riccio, A., Calori, G., D'Allura, A., and Gioli, B., Performance Analysis of Planetary Boundary Layer Parameterization Schemes in WRF Modelling Set Up over Southern Italy, Amosphere-Basel, *9*(7), 272; https://doi.org/10.3390/atmos9070272, 2018.

Worden, J.R., Bloom, A.A., Pandey, S., Jiang, Z., Worden, H.M., Walker, T.W., Houweling, S., and Röckmann, T, Reduced biomass burning emissions reconcile conflicting estimates of the post-2006 atmospheric methane budget, Nat. Commun., vol. 8, Article number: 2227, https://doi.org/10.1038/s41467-017-02246-0, 2017.

Xu, H., Wang, Y., and Wang, M., "The Performance of a Scale-Aware Non-local PBL Scheme for the Sub-kilometer Simulation of a Deep CBL over the Taklimakan Desert," Adv. Meteorol., vol. 2018, Article ID 8759594, 12 pages, https://doi.org/10.1155/2018/8759594, 2018.

**Authors' response to comments by anonymous referee #2 on Atmos. Chem. Phys. Discuss., manuscript acp-2018-948, "Country-scale greenhouse gases budgets using shipborne measurements: a case study for the United Kingdom and Ireland." by Helfter et al.**

We thank the referee for the positive assessment of our manuscript and for commending the rigour of the data analysis. We have addressed all of the comments and suggestions which were raised by the referee.

The referee's comments are set in bold, our responses in italics and new text, quoted from the revised manuscript, is highlighted in blue.

**Main comment**

**I did notice, that the background concentration of CO2 and CH4 had so much variability, and that the authors used a mathematical fitting routine to obtain a smoothened background signal. Based on previous studies using the mass balance approach, the large variability in the background actually contributes a significant variability (i.e. uncertainty) in the estimated fluxes.**

*We agree with the referee and we have added an uncertainty analysis to complement the temporal variability analysis initially presented in the manuscript and the sources of errors are also discussed in more depth.*

- ▪ *The text added to the discussion section is provided below (this also discusses other sources of uncertainty and bias):*

"Of the four main assumptions listed above, points c) and d) are the most subjective because they could not be verified nor quantified. Assumption a) (air mass travel from West to East) can be considered to be reasonably well-constrained owing to the data screening procedure at the pre-processing stage. Violations of the stationarity assumption (point b) due to significant changes in the mean PBL height at sub-hourly time step would either be captured, in part or entirely, during the next hourly averaging period, or go unnoticed in the case of very transient non-stationary events. Whilst the temporal variability of the mean PBL height for the spatial domain considered can be quantified and propagated through the emissions budgets calculations as measurement uncertainty, the potential bias between model output and observations is unknown. Recent studies have compared different WRF parametrisation schemes with observed PBL height and found that, in general, the YSU scheme used in this study performs reasonably well in terms of predicting PBL height with minimum bias typically observed before midday (Hu et al., 2010, Banks et al., 2016, Tyagi et al., 2018, Xu et al., 2018); however these studies also highlighted that model performance can vary significantly between sites and time of day, and that YSU tends to underestimate the PBL height over the sea (Tyagi et al., 2018). Comparisons between observations and model outputs of wind speed profiles for different parametrisation schemes also found substantial variability, both intra- and inter-model, with the YSU scheme exhibiting a tendency to overestimate wind speeds (Balzarini, 2014, Tyagi, 2018). The formation of sea breezes adds another level of complexity to the modelling of PBL height and wind speed, in particular in the southern North Sea where the orientation of the coastlines and their proximity to one another have been shown to induce sea breeze formation and to influence sea breeze type and offshore extent (Steele et al, 2013; Steele et al., 2014). Furthermore, not all WRF parametrisation schemes are equal in performance with respect to sea breeze conditions; recent studies show that the YSU scheme used here exhibited the smallest bias for wind speeds measured onshore under complex sea breeze conditions (Steele et al., 2014) and that it also captured the temporal evolution of the atmospheric boundary layer height better than other schemes (Salvador et al., 2016).

Intrinsic, unquantifiable biases on the mixing layer heights and mean wind speeds derived from the WRF model are hence likely. Wind speed and enhancement above background concentration were found to be to dominant uncertainty terms, jointly accounting for over 80% of the total uncertainty in all seasons (Table S2 of the Supplementary Material). In contrast, nudging the baseline concentrations measured at Mace Head by a time lag estimated from the mean air mass travel time had only a very modest impact on the final budgets (Table S2). The two measures of errors proposed in this paper (based on temporal variability and total uncertainty through error propagation) yield on the whole comparable results, with the main discrepancy found for the autumn budget (years used: 2015-2017) where the total uncertainty was almost four-fold the value obtained by considering the temporal variability alone. The autumn uncertainty was brought in line with the temporal variability estimate for both gases when the day/night weighting was applied. Whilst the variability and the total uncertainty are useful as first approximations for the confidence in the emission budgets, they should be treated as potential lower limits because of the unquantified bias between WRF model outputs and actual values of the PBL height and wind speed."

- *The uncertainty analysis and error propagation methodology added in the Methods section is given below (new text in blue):*

"Uncertainty and error propagation

In addition to the temporal variability $\Delta F_c$, (Eq. 4) we calculated the uncertainty on the total fluxes arising from the uncertainties on the individual terms of the mass balance equation. Noting that $dx$ represents the distance travelled by the ship with speed $v_{ship}$ during the infinitesimal time interval $dt$, Eq. 2 can be reformulated to express the partial flux $f_c$ through a 2-dimensional plane spanning the horizontal distance $dx$ as a function of $v_{ship}$ and $dt$ (Eq. 5).

$$f_C = \int_{z_{ground}}^{z_{PBL}} \Delta \chi_C . U . n_{air}(z) . \cos\theta . v_{ship} \; dt \, dz \tag{5}$$

Applying the rules of error propagation, the error on the flux term $f_c$ ($\delta f_c$) is given by (with $N_{air}$, the value of the integral of $n_{air}(z)$ evaluated over time step $dt$):

$$\frac{\delta f_c}{|f_c|} = \sqrt{\left(\frac{\delta \chi_c}{\chi_c}\right)^2 + \left(\frac{\delta U}{U}\right)^2 + \left(\frac{\delta \cos\theta}{\cos\theta}\right)^2 + \left(\frac{\delta v_{ship}}{v_{ship}}\right)^2 + \left(\frac{\delta dt}{dt}\right)^2 + \left(\frac{\delta \int_{z_{ground}}^{z_{PBL}} n_{air}(z) \; dz}{N_{air}}\right)^2} \tag{6}$$

Assuming that, (a) the uncertainty on $dt$ is negligible, and (b) the uncertainty on the PBL height ($z_{PBL}$) is the dominant error term in the integral of $n_{air}(z)$ between height $z_{ground}$ and $z_{PBL}$, Eq. 6 can be approximated as:

$$\frac{\delta f_c}{|f_c|} \approx \sqrt{\left(\frac{\delta \chi_c}{\chi_c}\right)^2 + \left(\frac{\delta U}{U}\right)^2 + \left(\frac{\delta \cos\theta}{\cos\theta}\right)^2 + \left(\frac{\delta v_{ship}}{v_{ship}}\right)^2 + \left[\frac{\left(n_{air}(z_{PBL}) - n_{air}(z_{ground})\right).\delta z_{PBL}}{N_{air}}\right]^2} \tag{7}$$

Finally, similarly to Eq. 4, the total error on the flux $F_c$ ($\delta F_c$) calculated for a complete transect of the ship between $x_{min}$ and $x_{max}$ is given by:

$$\frac{\delta F_c}{|F_c|} = \sqrt{\sum_i^N \left\{\left(\frac{\delta f_c}{|f_c|}\right)_i\right\}^2} \tag{8}$$

The standard deviations of the individual terms in Eq. 7, calculated for each 5-minute averaging period and averaged over each nominal latitude bin, were used as proxies for uncertainties."

*For disambiguation, the terminology used in the previous version of the manuscript was changed from "uncertainty" to "variability" throughout the document. In the revised manuscript, the term uncertainty denotes the error propagated using Eq. 7 and 8. Table 3 was updated and now provides both budget variability and uncertainty. Finally, the relative contribution of each uncertainty term was evaluated for each season and year of the study and provided in Table S2 of the Supplementary Material document (reproduced below).*

Table S2: Relative contribution of the individual uncertainty terms to the total uncertainty and total uncertainty on the calculated emissions budgets per season and year of the study. The difference between time-lagged and instantaneous emissions budgets illustrates the impact of factoring in the mean West-to-East air mass travel time in the selection of the reference concentrations measured at Mace Head.

| Season | Year | Relative contribution to total uncertainty [%] | | | | | Total uncertainty on emissions budget [%] | | Difference between time-lagged and instantaneous emissions budgets [%] | | Mean air mass travel time ± SD [hour] |
| | | Wind speed in PBL | Molar density | Mole fraction (enhancement above background) | Projection angle θ | Ship speed | $CO_2$ | $CH_4$ | $CO_2$ | $CH_4$ | |
|---|---|---|---|---|---|---|---|---|---|---|---|
| Winter | 2015 | - | - | - | - | - | - | - | - | - | 14.7 ± 4.7 |
| Spring | 2015 | 26 | 4 | 67 | 0 | 2 | 23 | 30 | 2.6 | 0.9 | 15.8 ± 5.0 |
| Summer | 2015 | 39 | 3 | 54 | 1 | 3 | 288 | 160 | 14.5 | 0.3 | 23.1 ± 9.9 |
| Autumn | 2015 | 48 | 5 | 43 | 2 | 2 | 74 | 11 | 2.0 | 0.3 | 15.8 ± 5.0 |
| Winter | 2016 | - | - | - | - | - | - | - | - | - | 15.2 ± 0.5 |
| Spring | 2016 | - | - | - | - | - | - | - | - | - | 14.7 ± 2.6 |
| Summer | 2016 | 45 | 4 | 49 | 1 | 2 | 63 | 64 | 1.4 | 0.5 | 20.2 ± 8.8 |
| Autumn | 2016 | 31 | 3 | 63 | 1 | 2 | 63 | 53 | 0.2 | 0.2 | 16.4 ± 7.4 |
| Winter | 2017 | 80 | 7 | 8 | 1 | 4 | 5 | 6 | 0.4 | 0.2 | 13.5 ± 4.1 |
| Spring | 2017 | 62 | 7 | 26 | 1 | 4 | 20 | 29 | 0.5 | 0.1 | 16.4 ± 6.2 |
| Summer | 2017 | 44 | 4 | 49 | 1 | 2 | 53 | 44 | 2.2 | 0.2 | 18.3 ± 6.4 |
| Autumn | 2017 | 71 | 4 | 20 | 1 | 4 | 3 | 4 | 0.9 | 0.2 | 15.9 ± 4.8 |

**Specific comments**

(1) **Not all your readers will be familiar with the geography of the measurement domain. Please provide a map of your measurement site together with the path of the ferry used in the study. Please label the map with the cities for reference. Direction of prevailing winds throughout the season will be also useful so that the reader can clearly see the transect of the ferry relative to the prevailing winds. This should be your figure 1. It helps if you set the stage for your readers:**

*We have added a map (new Fig. 1) which clarifies the locations of the Mace Head site, start and end points of the ferry route and other cities mentioned in the manuscript.*

[Figure]

**Figure 1.** Google Earth map centred on the United Kingdom and Ireland. The route of the ferry is indicated by a dark blue line joining the ports of Rosyth (Scotland, UK) and Zeebrugge (Belgium). The location of the Mace Head measurement station on the west coast of Ireland, which provided the carbon dioxide and methane concentration baselines, is indicated by a red star. The cities indicated by yellow stars are locations of interest cited in the discussion (Section 4).

*Seasonal wind roses were derived for each year of the study and were summarised in Figure S2 of the new Supplementary Material.*

[Figure]

Figure S2: Seasonal variability of the prevailing wind direction in the PBL for the three years of the study (2015-2017). The radial unit is the normalised frequency counts of the observations. Plot created with R-package openair (Carslaw and Ropkins, 2012).

5   **(2) Figure 1 already shows the Hysplit backward trajectory frequencies but the reader will just assume that the ship is located where the highest frequency is found (color red). Also, the two rows apparently show two succeeding days in May 2015. That is not clearly described in the figure caption and the reader discovers this only after staring at the figure. I think that this figure should not be your first figure.**

10  *The location of the ferry, although marked by a star-shaped symbol, was not clear in the original version of this figure due to the colour scheme used. The figure (now Figure 2 in the revised version of the manuscript) has been updated and the location of the ferry is indicated by an arrow.*

*The caption has been updated to clarify that the figure shows on single sailing which spanned two days.*

[Figure]

**Figure 2.** Backward trajectory frequencies for a South-bound sailing with westerly wind conditions (sailing start 17/05/2015 12:00, end 18/05/2015 10:00). The coloured contours represent the normalised frequency counts (number of end points in a 0.5°x 0.5° grid cell divided by the maximum number of end points in any grid cell, expressed as a percentage) and the source corresponds to the location of the ferry (indicated by an arrow). The trajectories were run backward for 24 hours at 3-hour intervals using GDAS 1-degree global meteorology (NOAA, 2018).

**(3) Page 3: You state that background measurements were taken at the Mace Head site in Ireland – all the more reason why your figure 1 should include what's stated above in (1) but also the location of the background site. It's important to set the stage for the reader for greater appreciation of the measurements and the analysis.**

*A new Figure 1 has been created as suggested the referee. This figure shows the location of the Mace Head background measurement site along with the other locations of interest referenced in the manuscript.*

**(4) Page 3: Your figure 1 should be something like your Figure 2 but with more detail such as an arrow that shows the wind direction. It is likely not westerly winds throughout the year. It will be instructive if the authors are deliberate about stating/describing the meteorological conditions throughout the year. It would be good to show a wind rose plot to support this.**

*We agree that this point is worth clarifying and we have therefore added seasonal wind roses for each year of the study into the new Supplementary Material document (Fig. S2). See copy of Fig. S2 under earlier comment.*

**(5) Page 3 line 16: Rosyth – Zeebrugge are two important ferry end points that should be included in Figure 1.**

*Rosyth and Zeebrugge are clearly marked in the new Figure 1.*

**(6) Page 3, line 30: Please state in the text where the calibration gases were obtained, how many was used, and their mixing ratios. Did they span the expected range of measured $CO_2$ and $CH_4$ mixing ratios from the measurement site? I note that the calibration gas was measured for 15 minutes? How often do you obtain a data point? Every 1 minute, every 15 seconds? Please state. How did you obtain the mean and uncertainty that was reported in Table 2, i.e. how many data points did you average every 15 minutes? What is the coverage factor k that was mentioned in Table 2? Please explain. And did you purge with the reference gas or with zero air?**

*For clarity, we added the following text on page 3, line 34 (new text in blue):*

"Calibrations using three gases spanning a realistic range of $CO_2$ and $CH_4$ concentrations ran every 169 hours and lasted 65 minutes in total. The references gases were calibrated by the Swiss Federal Laboratories for Materials Testing and Research (EMPA, Dübendorf, Switzerland) using a Picarro 1301 CRDS. The calibrations scales (NOAA/ ESRL) were WMO-CH$_4$-X2004 for methane and WMO-CO$_2$-X2007 for carbon dioxide. Each gas standard was measured at 1 Hz for 15 minutes and average and standard deviation were derived for the 15-minute period. A 5-minute purge period using the gas standard to be measured was observed before each active averaging period to flush out residual gas and eliminate sample contamination. Each calibration event ended with a 5-minute purge period using ambient air before resuming normal operations. The gas concentration time series were corrected using linear temporal interpolations between calibration events. Table 1 provides a list of observables; Table 2 summarises the weekly auto-calibration procedure and provides information on the three calibration gases used".

*Table 2 caused confusion regarding which calibration gas was used in each step of the sequence and we therefore a new column (calibration gas number) to clarify the reference gas usage. The revised Table 2 and caption are now (new text in blue clarifying the meaning of the coverage factor k):*

**Table 2.** Details of the weekly auto-calibration sequence (interval 169 hours) and reference gases. The references gases were calibrated by the Swiss Federal Laboratories for Materials Testing and Research (EMPA, Dübendorf, Switzerland) using a Picarro 1301 CRDS. Calibrations scales (NOAA/ ESRL): WMO-CH$_4$-X2004 for methane and WMO-CO$_2$-X2007 for carbon dioxide. The measurement uncertainties correspond to the standard uncertainty multiplied by a coverage factor k = 2, which provides a level of confidence of approximately 95 %.

| Step | Type | Time interval [s] | Calibration standard number | CO$_2$ ± uncertainty [ppm] | CH$_4$ ± uncertainty [ppb] |
|------|------|------|------|------|------|
| 1 | Purge | 300 | 1 | 384.23 ± 0.15 | 1815.36 ± 1.45 |
| 2 | Measurement | 900 | 1 | 384.23 ± 0.15 | 1815.36 ± 1.45 |
| 3 | Purge | 300 | 2 | 418.29 ± 0.16 | 2018.06 ± 1.58 |
| 4 | Measurement | 900 | 2 | 418.29 ± 0.16 | 2018.06 ± 1.58 |
| 5 | Purge | 300 | 3 | 474.86 ± 0.18 | 2426.77 ± 1.86 |
| 6 | Measurement | 900 | 3 | 474.86 ± 0.18 | 2426.77 ± 1.86 |
| 7 | Purge | 300 | Ambient air | Ambient air | Ambient air |

**(7) Page 4 and page 5: Assumptions: What is tricky about the mass balance approach is the choice/estimation of the background. It can contribute one of the largest uncertainty in the estimation of the emission flux – because the air mass travels a couple of hours from the background site to the receptor site. Your measurements at the Mace Head site showed significant variability in the background. What's the effect of this variability on the estimated flux?**

▪ **It is possible that the uncertainty is significantly larger than what equation (4) is estimating simply because there is so much variability in the background mixing ratio of CO2 and CH4. Based on the results of previous studies, the variability in the background significantly contributes to the uncertainty in the estimated fluxes. The authors are then advised to do a sensitivity analysis of the obtained fluxes using the standard deviations in the background obtained from Mace Head and comment on the results in the discussion (comparing against the uncertainty obtained in equation 4).**

*In addition to the measure of the temporal variability presented in the first version of the manuscript, we have also estimated the total uncertainty on the calculated budgets which arise from the uncertainties on the individual terms contributing to the mass balance. The uncertainty calculation and error propagation is described in Section 2.2.4 (Mass balance calculations) and has already been covered in this document under an earlier comment.*

*Table 3 has been revised and both variability and uncertainty values are provided for each budget term.*

**Table 3.** Seasonal and annual budgets for $CO_2$ and $CH_4$ for the United Kingdom (excluding Scotland) and Ireland estimated by a mass balance approach using concentrations measured at the Mace Head station (Republic of Ireland; 53° 19' 19.2'' N, 9° 54' 3.599'' W) and on board the freight ferry which serves the Rosyth (Scotland, UK; 56° 1' 21.611'' N, 3° 26' 21.558'' W) to Zeebrugge (Belgium; 51° 21' 16.96'' N, 3° 10' 34.645'' E) route. Seasonal budgets were calculated by year – where sufficient data was available; seasonal budgets were also derived using the entire dataset with and without segregation of the raw fluxes into day and night components. Annual budgets were calculated with and without seasonality and with and without day/night segregation. The variability and uncertainty terms were calculated using Eq. 4 and Eq. 7-8, respectively.

| Season | Year | Flux ± uncertainty (variability) [Tg] | |
|---|---|---|---|
| | | $CO_2$ | $CH_4$ |
| Winter | 2015 | - | - |
| Spring | 2015 | 92.6 ± 21.1 (34.7) | 0.43 ± 0.13 (0.11) |
| Summer | 2015 | 27.6 ± 79.5 (46.8) | 0.45 ± 0.72 (0.09) |
| Autumn | 2015 | 286.4 ± 35.4 (47.6) | 0.61 ± 0.07 (0.14) |
| Winter | 2016 | - | - |
| Spring | 2016 | - | - |
| Summer | 2016 | 131.6 ± 82.6 (36.5) | 0.39 ± 0.25 (0.09) |
| Autumn | 2016 | 261.3 ± 164.3 (56.4) | 0.75 ± 0.40 (0.16) |
| Winter | 2017 | 341 ± 17.2 (62.1) | 0.78 ± 0.05 (0.38) |
| Spring | 2017 | 197.5 ± 40.4 (27.9) | 0.49 ± 0.14 (0.07) |
| Summer | 2017 | 155 ± 81.8 (77.6) | 0.32 ± 0.14 (0.06) |
| Autumn | 2017 | 363.4 ± 12.1 (65.7) | 1.03 ± 0.04 (0.15) |
| Winter | 2016 & 2017 | 379.1 ± 26.6 (68.8) | 0.89 ± 0.08 (0.35) |
| Spring | 2015 – 2017 | 161.5 ± 30.9 (41.2) | 0.55 ± 0.08 (0.17) |
| Summer | 2015 – 2017 | 123.6 ± 76.9 (64.6) | 0.38 ± 0.25 (0.09) |
| Autumn | 2015 – 2017 | 250.2 ± 200.1 (57.8) | 0.72 ± 0.40 (0.16) |
| Winter (day/night weighting) | 2016 & 2017 | 357.8 ± 26.2 (66.8) | 0.82 ± 0.08 (0.34) |
| Spring (day/night weighting) | 2015 – 2017 | 162.5 ± 30.9 (55.0) | 0.57 ± 0.08 (0.22) |
| Summer (day/night weighting) | 2015 – 2017 | 127.7 ± 76.9 (78.7) | 0.39 ± 0.25 (0.12) |
| Autumn (day/night weighting) | 2015 – 2017 | 232.9 ± 57.8 (72.2) | 0.67 ± 0.16 (0.19) |
| Annual (from seasonal budgets) | 2015 – 2017 | 914.4 ± 218.1 (118.1) | 2.55 ± 0.48 (0.43) |
| Annual (from seasonal, day/night weighted budgets) | 2015 – 2017 | 881.0 ± 125.8 (137.5) | 2.44 ± 0.30 (0.47) |
| Annual (no seasons) | 2015 – 2017 | 708.3 ± 270.4 (241.9) | 2.1 ± 0.67 (0.63) |
| Annual (no seasons, day/night weighted) | 2015 – 2017 | 598.3 ± 250.1 (274.9) | 1.66 ± 0.60 (0.94) |

| | | | |
|---|---|---|---|
| UK (Department for Business, 2017) | 2015 | 415.1 | 2.1 |
| RoI (Agency, 2017) | 2015 | 38.4 | 0.53 |
| Scotland (Inventory, 2018) | 2015 | 30.8 | 0.34 |
| Total inventory (UK – Scotland + RoI) | 2015 | 422.7 | 2.29 |
| Ganesan (Ganesan et al., 2015) | 2012 - 2014 | - | 1.65 - 2.67 |
| Bergamaschi (Bergamaschi et al., 2015) | 2006 - 2007 | | 3.1 – 3.5 |

*In addition, the relative contributions to the total uncertainty of the individual terms used to calculate the mass balance budgets have been evaluated and are presented in Table S2 of the new Supplementary Material (also presented in this document in response to an earlier comment). This shows that the dominant uncertainty terms are the mean wind speed in the PBL and the*
5  *mole fraction enhancement above background. In contrast, applying a time lag to fix the value of the mole fraction baselines with respect to the estimated mean West-East air mass travel time only has a very moderate impact on the budgets (Table S2). Uncertainty and variability are also discussed in Section 4 and the new text has already been reproduced in this document under an earlier comment.*

     ▪  **Furthermore, on page 5 regarding the mass balance approach, what was the time – shift interval that was used**
10     **to obtain the background? I ask this question because there is a significant travel time for the air mass to reach the receptor site. This is also the reason why in previous aircraft-based mass balance approaches, the mixing ratios at the "wings" of the transect (outside the plume) were actually used as the baseline or background mixing ratio.**

*The sentence was expanded to clarify this point (see below, new text in blue). The impact of time-shifting the baseline on the*
15  *total uncertainty was also discussed in Section 4 (see response to comment above).*

"The baseline mole fractions used to calculate the upwind enhancement of compound $c$ were time-shifted in order to account for the mean air mass travel time across the domain (time taken to travel West-East from the longitude of the Mace Head station to the location of the ferry at hourly mean wind speed derived from the WRF model; see Table S2 of the Supplementary material for seasonal mean values and standard deviations)."

20  *Values of the mean seasonal air mass travel time and their impact on the final budgets are summarised in Table S2 of the new Supplementary Material. This table is available on page 4 of this document.*

     (8)  **Page 4 on Data screening: Please state in this section that the histogram of data points showed in Figure 3 correspond to the data points that satisfy the data screening protocol, which emphasizes that only the data with westerly winds were used in the averaging and flux calculations. Out of all the data points that you**
25     **collected, how many points were used in the analysis? What's the percentage of useful data? Are there more points for certain months relative to others?**

*We added the following sentences at the end of section 2.2.1 on Data screening (the new Table S1 is also referenced):*

"The temporal coverage of the data points which satisfied the criteria listed above is presented in histogram form in Fig. 4.
30  The full details of the data availability for the study period 2015-2017 are summarised in table S1 of the Supplementary Material."

Table S1: Data availability per season and year of the study. The total number of measured points and the number of points which satisfied the data screening criteria are given.

| Year | Season | Total number of points | Number of quality-controlled points | Proportion of quality-controlled points [%] |
|---|---|---|---|---|
| 2015 | Winter | 5621 | 252 | 4 |
| 2015 | Spring | 7265 | 502 | 7 |
| 2015 | Summer | 12919 | 1232 | 9 |
| 2015 | Autumn | 7803 | 493 | 6 |
| 2015 | All seasons | 33608 | 2479 | 7 |
| 2016 | Winter | 4198 | 21 | 0.5 |
| 2016 | Spring | 9689 | 226 | 2 |
| 2016 | Summer | 9398 | 1650 | 18 |
| 2016 | Autumn | 20243 | 1618 | 8 |
| 2016 | All seasons | 43528 | 3515 | 8 |
| 2017 | Winter | 3228 | 618 | 17 |
| 2017 | Spring | 14459 | 1194 | 8 |
| 2017 | Summer | 15629 | 1447 | 9 |
| 2017 | Autumn | 3030 | 272 | 10 |
| 2017 | All seasons | 36746 | 3531 | 10 |
| All | Winter | 13447 | 891 | 6 |
| All | Spring | 31413 | 1922 | 6 |
| All | Summer | 37946 | 4329 | 11 |
| All | Autumn | 31076 | 2383 | 8 |
| All | All seasons | 113882 | 9525 | 8 |

**(9) Page 5:**
- **The authors used the wind speed and wind direction from the WRF model to calculate the fluxes. I am sure that there are multiple synoptic stations in UK and Ireland? How come the wind speed and wind direction data from those synoptic stations were not used in this study?**

*There is indeed a large number of synoptic stations in the UK but the decision to use wind speed data extracted from the WRF model was motivated by the use of the mean values within the PBL in the simplified formulation of the mass balance equation (Eq. 3).*

- **Were the back trajectories consistent with the synoptic station data?**

*We found that our HYSPLIT quality flags and the wind directions obtained at Mace Head and filtered for Westerly flow (240° – 300°) agreed in 70% of all cases over the 3-year study period. Cases where there were discrepancies between the HYSPLIT back trajectories and the wind direction filter were not included in the analysis because the data screening protocol was constructed as a logical AND condition.*

- **At what height above the ground were the back trajectories modelled for?**

*The back trajectories were modelled for a height of 500 m a.g.l. This information was added to the section on Data screening:*

*"72-hour back-trajectories (500 m a.g.l) for the Mace Head site and one point along route of the ferry (54.548 °N, 0.233 °W) as calculated with HYSPLIT (NOAA Air Resources Laboratory, 2018) exhibited air flow patterns inconsistent with the mass balance assumptions (i.e. non-westerly flow, evidence of re-circulation). "*

- **It will be good to show the time series of the PBL heights in the supplementary information for the domain and for the period of measurement.**

*Figure S1 of the Supplementary Material presents the daily mean PBL heights and standard deviations derived by averaging the hourly values obtained from the WRF model for the spatial domain considered.*

[Figure]

5  Figure S1: Daily mean PBL height (solid dots) and standard deviation (shaded ribbon) obtained by averaging the hourly values extracted from the WRF model with YSU scheme for the study period 2015-2017.

▪ **It would be also good to report if the modelled PBL depth has been validated with previous measurements (at least for those previous years when measurements of PBL depths were available), just to reassure**
10  **yourselves that the modelled values are sufficient to be used in your calculations.**

*The modelled PBL data could not be validated against measured values. PBL height and mean wind speed are treated as observables with known variability but with the caveat that their bias is unknown. This and the performance of WRF model are discussed explicitly in the revised version of Section 4 (see Page 1 and 2 of this document).*
15
**(10) Results.**
▪ **On seasonal and annual fluxes. Please check the units of your fluxes in Figure 6 and 7. I believe you meant g s-1 m-2 rather than g s-1 m-1.**

*Figures 6 & 7: The unit of $(g.s^{-1}).m^{-1}$ used in the manuscript is correct; as explained in the captions, the fluxes are integrated*
20  *over the height of the boundary layer height and are expressed in units of mass flux per meter travelled crosswind within each latitude bin per unit time.*

▪ **Report the total number of data points per season.**

*This information has been supplied in Table S1 of the new Supplementary Material (reproduced below).*

Table S1: Data availability per season and year of the study. The total number of measured points and the number of points
25  which satisfied the data screening criteria are given.

| Year | Season | Total number of points | Number of quality-controlled points | Proportion of quality-controlled points [%] |
|---|---|---|---|---|
| 2015 | Winter | 5621 | 252 | 4 |
| 2015 | Spring | 7265 | 502 | 7 |
| 2015 | Summer | 12919 | 1232 | 9 |
| 2015 | Autumn | 7803 | 493 | 6 |
| 2015 | All seasons | 33608 | 2479 | 7 |
| 2016 | Winter | 4198 | 21 | 0.5 |

| 2016 | Spring | 9689 | 226 | 2 |
|---|---|---|---|---|
| 2016 | Summer | 9398 | 1650 | 18 |
| 2016 | Autumn | 20243 | 1618 | 8 |
| 2016 | All seasons | 43528 | 3515 | 8 |
| 2017 | Winter | 3228 | 618 | 17 |
| 2017 | Spring | 14459 | 1194 | 8 |
| 2017 | Summer | 15629 | 1447 | 9 |
| 2017 | Autumn | 3030 | 272 | 10 |
| 2017 | All seasons | 36746 | 3531 | 10 |
| All | Winter | 13447 | 891 | 6 |
| All | Spring | 31413 | 1922 | 6 |
| All | Summer | 37946 | 4329 | 11 |
| All | Autumn | 31076 | 2383 | 8 |
| All | All seasons | 113882 | 9525 | 8 |

(11) **Figure 10 and 11**.

- What does the grey shade represent in the figures? Please explain in the figure caption.

*The following text was appended to the captions of Fig. 10-12 to clarify this point:* "The shaded area represents the 95% confidence interval of the linear regression.*"*

- **In Figure 11, you used the mean air temperature over the measurement domain. How many synoptic stations were used when you averaged the air temperature?**

*The UK Met Office reports the use of ca. 250 synoptic stations. This information has been added into the caption of Fig. 11 (see new text in blue below):*

"Seasonal budgets of $CH_4$ and $CO_2$ as function of mean UK air temperature derived from ca. 250 synoptic stations (source: Met Office, 2018). The shaded area represents the 95% confidence interval of the linear regression.*"*

**New references**

Balzarini, A., Angelini, F., Ferrero, L., Moscatelli, M., Perrone, M. G., Pirovano, G., Riva, G.M., Sangiorgi, G., Toppetti, A.M., Gobbi, G.P., and Bolzacchini, E., Sensitivity analysis of PBL schemes by comparing WRF model and experimental data, Geosci. Model Dev. Discuss., 7, 6133–6171, www.geosci-model-dev-discuss.net/7/6133/2014/, doi:10.5194/gmdd-7-6133-2014, 2014.

Banks R.F., Tiana-Alsina J., Baldasano J.M., Rocadenbosch F., Papayannis A., Solomos S., Tzanis C.G., Sensitivity of boundary-layer variables to PBL schemes in the WRF model based on surface meteorological observations, LIDAR, and radiosondes during the HygrA-CD campaign, Atmos. Res., Vol. 176–177, pp. 185-201, https://doi.org/10.1016/j.atmosres.2016.02.024, 2016.

BEIS (Department for Business, Energy and Industrial Strategy): UK Emission Mapping Methodology, available at https://uk-air.defra.gov.uk/assets/documents/reports/cat07/1812061112_MappingMethodology-for-NAEI-2016.pdf (last access 23 January 2019), 2016.

Carslaw, D.C. and K. Ropkins, (2012) openair — an R package for air quality data analysis. Environ. Modell. Softw., Vol. 27-28, 52-61.

[revised manuscript text omitted]

---

## Author Response (AR2)

**Authors' response to Co-Editor's comments on Atmos. Chem. Phys. Discuss., manuscript acp-2018-948, "Country-scale greenhouse gases budgets using shipborne measurements: a case study for the United Kingdom and Ireland." by Helfter et al.**

5   We thank the co-editor for approving the corrections and responses given to the referees' comments and for the very positive assessment of our manuscript. The comments and minor revisions requested by the co-editor and our responses thereto are given below (the co-editor's comments are set in bold, our responses in italics and new text, quoted from the revised manuscript is highlighted in blue). The revised version of the manuscript (with tracked changes) is provided after the responses to comments section.

**Question:**

- **It is still not entirely clear to me how PBL height and wind speed from WRF were used in the mass balance calculations. Section 2.2.3 mentions that spatial means of PBL height and wind speed were calculated for each hour, but it is not very clearly stated, how these were used in the mass balance calculations described in Sect**
15      **2.2.4. Am I right that for each 5-minute averaging interval (say 11:45-11:50 UTC), mean PBL height and wind speed from the corresponding hour (e.g. hourly mean of 11-12 UTC, or maybe from the closest hour, i.e. 12 UTC in the above case) were used in the calculations? Or did you apply a time shift, similar to the one applied to the Mace Head background?**

20   *Section 2.2.3 was expanded to clarify this point as follows:*

"Time series of hourly averages of wind speed, wind direction and PBL height were constructed for the data period 01/01/2014 to 31/12/2017. These hourly values were extrapolated to the 5-minute concentrations and ancillary (e.g. meteorological, ship speed, coordinates) time series by assigning PBL height and wind speed to the corresponding hour in the 5-minute dataset (e.g. the mean PBL height value estimated for the time period 3:00 - 4:00 UTC on 13/06/2015 was assigned to all 5-minute averaging
25   intervals from 3:00 to 3:55 UTC).

Daily means and standard deviations obtained by averaging the hourly values of the PBL heights derived from WRF for the study period 2015-2017 are presented in Fig. S1 of the Supplementary Material."

- **If the travel time from east to west was X hours, why did you compute mean winds and PBL heights for the**
30      **present hour h rather than for hour h-X/2, i.e. for the middle point of the air mass transit across the UK? Or maybe you did? Due to the potentially large diurnal variability of PBL heights, the difference between using hour h and using hour h-X/2 could be quite significant. Wouldn't it be useful to consider this in the uncertainty analysis?**

35   *This is a very pertinent point which was considered at the time of writing the original version of the manuscript. We arbitrarily chose not to apply a time shift to the PBL height and mean wind speed values because the budgets derived using hour h and hour h-X/2 (using the editor's notations) were not statistically different from one another. The table below provides examples of annual budgets and uncertainties calculated with and without time-shifting by X/2 to illustrate this statement.*

[revised manuscript text omitted]